# Study on the Anti-Ulcerative Colitis Effect of Pseudo-Ginsenoside RT4 Based on Gut Microbiota, Pharmacokinetics, and Tissue Distribution

**DOI:** 10.3390/ijms25020835

**Published:** 2024-01-09

**Authors:** Hui Yu, Caixia Wang, Junzhe Wu, Qianyun Wang, Hanlin Liu, Zhuoqiao Li, Shanmei He, Cuizhu Wang, Jinping Liu

**Affiliations:** 1School of Pharmaceutical Sciences, Jilin University, Changchun 130021, China; yuhui21@mails.jlu.edu.cn (H.Y.); cxwang20@mails.jlu.edu.cn (C.W.); wujz21@mails.jlu.edu.cn (J.W.); wqy22@mails.jlu.edu.cn (Q.W.); hanji23@mails.jlu.edu.cn (H.L.); lizq21@mails.jlu.edu.cn (Z.L.); hesm21@mails.jlu.edu.cn (S.H.); wangcuizhu@jlu.edu.cn (C.W.); 2Research Center of Natural Drug, Jilin University, Changchun 130021, China

**Keywords:** RT4, ulcerative colitis, intestinal flora, pharmacokinetics, tissue distribution

## Abstract

The purpose of this study was to explore the therapeutic effect of the oral administration of pseudo-ginsenoside RT4 (RT4) on ulcerative colitis (UC), and to determine the rate of absorption and distribution of RT4 in mice with UC. Balb/c mice were induced using dextran sulfate sodium salts (DSS) to establish the UC model, and 10, 20, or 40 mg/kg of RT4 was subsequently administered via gavage. The clinical symptoms, inflammatory response, intestinal barrier, content of total short-chain fatty acids (SCFAs), and gut microbiota were investigated. Caco-2 cells were induced to establish the epithelial barrier damage model using LPS, and an intervention was performed using 4, 8, and 16 µg/mL of RT4. The inflammatory factors, transient electrical resistance (TEER), and tight-junction protein expression were determined. Finally, pharmacokinetic and tissue distribution studies following the intragastric administration of RT4 in UC mice were performed. According to the results in mice, RT4 decreased the disease activity index (DAI) score, restored the colon length, reduced the levels of pro-inflammatory cytokines (TNF-*α*, IL-6, and IL-1*β*), and boosted the levels of immunosuppressive cytokine IL-10, increased the content of SCFAs, improved the colonic histopathology, maintained the ultrastructure of colonic mucosal epithelial cells, and corrected disturbances in the intestinal microbiota. Based on the results in caco-2 cells, RT4 reduced the levels of TNF-*α*, IL-6, and IL-1*β*; protected integrity of monolayers; and increased tight-junction protein expression. Additionally, the main pharmacokinetic parameters (C_max_, *T*_max_, *t*_1/2_, *V*_d_, *CL*, *AUC*) were obtained, the absolute bioavailability was calculated as 18.90% ± 2.70%, and the main distribution tissues were the small intestine and colon. In conclusion, RT4, with the features of slow elimination and directional distribution, could alleviate UC by inhibiting inflammatory factors, repairing the intestinal mucosal barrier, boosting the dominant intestinal microflora, and modulating the expression of SCFAs.

## 1. Introduction

Ulcerative colitis (UC) is an intractable intestinal inflammatory disease with a growing global morbidity rate [1,2]. The clinical symptoms include diarrhea, abdominal pain, bloody stools, and rectal bleeding. More seriously, long-term inflammatory injury can even progress to intestinal cancer [3]. Glucocorticoids, thiopurines, corticosteroids, and methotrexate have been used to treat UC therapeutically due to their anti-inflammatory or immunomodulatory effects [1]. However, the possibility exists that these drugs could, to some degree, stimulate the gastrointestinal tract and thus cause damage to the gastrointestinal mucosa. This inevitably could result in adverse reactions in particular with long-time use [4]. Due to their widespread distribution within the body, these drugs may even cause some side effects on bone marrow [5,6], blood [7], skin [8], and other tissues [9]. Consequently, to meet the clinical requirements, some therapies such as repair of the intestinal mucosal barrier, balancing the gut microbiota, or specific and targeted distribution are usually considered as the strategy for new courses of treatment. To mention a few examples, M10, a naturally occurring flavonoid of myricetin derivatized with the properties of colon-specific distribution, was reported as treating UC mice by modifying the composition of the gut microbiota [10]. Puerarin, a natural polyphenol product, may relieve UC via enhancing mucin secretion, regulating mucin-utilizing bacteria, and increasing levels of SCFAs in mice [11]. Artesunate, a diterpenoid lactone, has also shown positive anti-UC effects through significantly protecting the integrity of the intestinal barrier and reducing inflammatory cell infiltration in UC mice [12]. The ginsenoside Rh1, a tetracyclic triterpenoid glycoside, has also shown regulatory activity in relation to the crosstalk between macrophages and gut microbiota in UC mice [13].

The pseudo-ginsenoside RT4 (RT4, CAS: 98474-77-2), belonging to the tetracyclic triterpenoids, naturally exists in *Panax pseudoginseng* subsp. *Himalaicus* [14], *Panax vietnamensis* Ha et Grushv [15], and *Panax quinquefolius* L. [16]. It has been reported that RT4 shows both a hepatocytoprotective effect [17] and an anti-inflammatory effect [18,19]. It is worth mentioning that intraperitoneal injections of RT4 have the potential to alleviate disease symptoms, increase colon length and weight, and improve the pathological state of colonic tissues in UC mice [20]. It has further been revealed that RT4 could down-regulate the levels of pro-inflammatory cytokines (IL-1*β*, TNF-*α*) and up-regulate the level of anti-inflammatory cytokines (IL-10) in the colonic tissue of UC mice. It has also been discovered that RT4 has the same effect on inflammatory factors in the RAW 264.7 cell inflammation model constructed with lipopolysaccharides (LPS) [20]. However, compared to intra-peritoneal injections, oral administration is more convenient and shows a better patient compliance with the drug regimen. Subsequently, there are some important issues that need further in-depth research, for example, the effect of the oral administration of RT4 on UC remains still unclear, the effects of RT4 on intestinal mucosal barrier and on gut microbiota are also ambiguous, and the oral pharmacokinetics and distribution of RT4 in the colon remain to be clarified.

In the present study, first, both DSS-induced UC mice and LPS-induced epithelial barrier damage caco-2 cells were used to explore the therapeutic effect of the oral administration of RT4 on UC. Disease symptoms, inflammatory cytokines, intestine mucosal barrier function, short-chain fatty acids (SCFAs), and gut microbiota were all observed or detected. Second, the oral pharmacokinetics and distribution of RT4 in UC mice were also investigated.

## 2. Results

### 2.1. Effects on DSS-Induced UC Mice

#### 2.1.1. DAI Score, Body Weight, Colon Length, and Spleen Index

To investigate the therapeutic effect of RT4, a UC model was established by administrating 3% dextran sulfate sodium salt (DSS) for 1 week. Then, the mice were treated with RT4 at three dosages for 3 days via gavage, and the standard treatment, sulfasalazine (SASP, a positive control drug), was included in the experiment to allow comparisons to be drawn between the test compound and the established drug treatment.

As shown in Figure 1A,B, during the 7-day model-building period, the weight of the normal group showed a continuous weight gain, and the disease activity index (DAI) score was almost zero. From Day 8 to the end of the experimental period, compared with the normal group, the body weight of the model group significantly decreased over time, and the DAI score significantly increased (*p* < 0.05). However, the colitis symptoms (DAI score increase and body weight loss) of the RT4 groups were significantly alleviated to varying degrees. Compared with the model group, the middle dosage and high dosage of RT4 could significantly reduce the DAI scores (*p* < 0.05). Moreover, a high dosage of RT4 could obviously increase body weight compared with the model group (*p* < 0.05).

Colon length is negatively related to the severity of colitis. As shown in Figure 1C,E, the colons of mice in the model group were significantly shortened compared to the normal group (*p* < 0.001), indicating that a severe inflammatory reaction occurred in the colonic tissues. Compared with the model group, the colon length of the RT4-M or RT4-H groups was obviously lengthened (*p* < 0.01, *p* < 0.001), suggesting the inflammation had been alleviated. However, although the colon length of the RT4-L group was longer than that of the model group, there was no statistical significance (*p* > 0.05).

The degree of splenomegaly was also positively correlated with the severity of colitis. The higher the degree of splenomegaly, the more extensive the development of colitis. The increase in spleen index, which reflects changes in the spleen, indicated the presence of congestion, edema, or hypertrophy in the spleen. In the current experiment, we observed that the spleen index apparently increased in the model group compared to the normal group (*p* < 0.001), implying that severe splenomegaly had occurred. After the intervention with RT4, the spleen index was dose-dependently lowered compared to the model group (*p* < 0.05, *p* < 0.01, *p* < 0.001) (Figure 1D); therefore, it was demonstrated that RT4 could relieve splenomegaly in UC mice.

In particular, the impact on body weight, DAI score, colon length, or spleen index of a high dosage of RT4 were all similar to the effect of SASP (*p* > 0.05).

#### 2.1.2. Systematic Inflammation

Inflammatory cytokines are involved in the pathogenesis of colitis. To further clarify the anti-UC activity of RT4, the levels of TNF-*α*, IL-6, IL-1*β* (the pro-inflammatory factors), and IL-10 (the anti-inflammatory factor) in serum or the colon were examined. According to the determination results, the DSS treatment visibly increased the expression of three pro-inflammatory factors, and decreased the expression of anti-inflammatory factors compared with the normal group (Figure 2A–D, *p* < 0.001). Meanwhile, these inflammatory cytokines were ameliorated to various degrees through treatment with RT4 or SASP. Compared with the model group, RT4-H markedly decreased the levels of three pro-inflammatory factors both in serum and in the colon (all *p* < 0.01), and obviously increased the levels of anti-inflammatory factor in the colon or in serum (*p* < 0.01 or *p* < 0.05). As for the RT4-M group, the levels of TNF-*α* (*p* < 0.01), IL-6 (*p* < 0.01, *p* < 0.05), and IL-1*β* (*p* < 0.01) in the colon or in serum were effectively decreased, and the level of IL-10 (*p* < 0.05) in the colon was markedly increased; meanwhile, RT4-L decreased the levels of three pro-inflammatory factors in the colon (*p* < 0.01, *p* < 0.01, *p* < 0.05). In addition, there was almost no statistically significant difference (*p* > 0.05) between the SASP group and RT4-H group in the regulation of IL-6 and IL-1*β* in the colon.

#### 2.1.3. Assay of Total Short-Chain Fatty Acids’ (SCFAs) Concentration

SCFAs, the microbial metabolites in the intestine, are the crucial energy sources for intestinal epithelial cells. SCFAs could modulate the intestinal mucosal barrier function. To evaluate the effects of RT4 on SCFAs, the levels of total SCFAs in cecal content were determined. Just as Figure 2E shows, compared to the normal group, the content of total SCFAs in the DSS-induced model group was effectively decreased (*p* < 0.01), while RT4 treatment could markedly restore the contents of total SCFAs toward normal levels (*p* < 0.01). Therefore, it was speculated that the improvement in colonic mucosal barrier function by RT4 in UC mice might be related to the increase in SCFAs content. And the effect of RT4-H was similar to the effect of SASP with no significant statistical difference (*p* > 0.05).

#### 2.1.4. Histopathological Examination

To investigate the impact of the RT4 oral formulation on colonic histopathology, we conducted the histopathological examination. As shown in Figure 3A, the colonic tissue of normal mice maintained its structural clearness and integrity, with abundant and tight glands, and there was no obvious inflammatory cell infiltration. However, the colonic structures of UC model mice were severely damaged, the crypt structure had disappeared, and a significant number of inflammatory cells had appeared. Compared with the model group, the groups provided with three dosages of RT4 gradually improved the pathological features in the colon and the inflammatory cell infiltration was reduced. The RT4-L group showed a partial restoration of the structure of colonic crypts and the inflammatory infiltration in the colon was demonstrably alleviated. As for the RT4-M group, there was a clear restoration of both the structure and quantity of crypts within the colon, and recovery from the inflammatory infiltration was also illustrated. In the RT4-H group and SASP group, the structure and quantity of crypts tended to be normal, there was a small amount of inflammatory cell infiltration, and the colonic structure was mainly restored as clear and intact.

#### 2.1.5. Transmission Electron Microscopy (TEM) Analysis

In order to provide detailed insights into the particular state of epithelial cells of colon tissue, we carried out TEM experiments (Figure 3B). The microvilli of colonic mucosal cells in the normal group were slender, dense, and well arranged. The tight junctions between cells were regular, the organelles were not denatured, and there was no vacuole change in cytoplasm. The microvilli of the model group mice were sparse, shortened, or had even disappeared, with slack intercellular connections, uneven cytoplasm, and vacuolar degeneration of the mitochondria. Compared with the model group, RT4 could restore the integrity of intestinal mucosal cells and the orderly arrangement of microvilli in a dose-dependent manner. In particular, there was almost no degeneration of organelles in the RT4-H group, and the ultrastructure of the colonic mucosal epithelial cells was well maintained, which were similar to the effects in the SASP group.

#### 2.1.6. RT4 Modulated the Composition of Gut Microbiota

Recently, growing evidence has shown that an imbalance of the intestinal flora is a critical pathogenic factor in colitis [21]. To elucidate whether the beneficial effects of RT4 on colitis are related to the regulation of intestinal microbiota, we collected fresh feces from the mice and performed 16S rRNA sequencing detection.

Both *α*-diversity (rarefaction curve and Shannon index) and *β*-diversity (principal coordinate analysis, PcoA) analyses were performed to investigate the gut microbiota in the normal, model, and RT4-H groups. Firstly, the rarefaction curve of the intestinal bacterial community was established to ascertain whether the sequencing depth of gut microbiota in each sample was sufficiently captured. As shown in Figure 4A, the operational taxonomic unit (OTU) level tended to plateau with the increase in the number of reads/samples, revealing that all samples had reached significant depths and proving the reliability of the sequencing results. As shown in Figure 4B, 12,614 OTUs with a similarity greater than 97% were obtained from 30 samples, the number (4677) of OTUs in the model group was relatively decreased compared to the normal group (8238 OTUs), while the high doses in the RT4 group (7228 OTUs) increased the number of OTUs. Secondly, the Shannon index was used to reflect species diversity [22]. The Shannon index of the model group (Figure 4C) was significantly lower than that of the normal group (*p* < 0.05), indicating a poor microbial community diversity in UC mice. However, in the RT4 group, the intestinal bacterial diversity was restored and the bacterial composition was close to that of normal mice (*p* > 0.05). Thirdly, the similarities or differences in community composition and structure among different samples could be compared via *β*-diversity analysis. The PcoA results (Figure 4D) indicated that the DSS-stimulated colitis had caused a structural change in the intestinal microbiota. At the same time, the RT4 group was close to the normal group, indicating that the improvements in the intestinal microbiota structure contributed to the alleviating effect of RT4 on UC.

In summary, the above results suggested that the effective treatment of UC mice with RT4 was most likely related to the enhancement in the gut microbiota structure.

Aiming at revealing the detailed composition of the gut microbiome, differential analysis was conducted at the phylum and genus levels. At the phylum level (Figure 5A), compared to the normal group, the occurrence of colitis resulted in an increase of 83.79 times, 13.54 times, 13.49 times, 1.13 times, and 1.92 times in the abundance of *Acidobacteria*, *Proteobacteria*, *Epsilonbacteraeota*, and *Actinobacteria*, respectively; meanwhile the abundance of *Tenericutes* and *Bacteroidetes* was 6.32 times and 1.79 times lower, respectively. Compared to the model group, the intervention with RT4 recovered the abundance of *Acidobacteria* (from 0.001% to 0.076%), *Proteobacteria* (from 27.335% to 2.987%), *Epsilonbacteraeota* (from 2.843% to 0.468%), *Actinobacteria* (from 0.943% to 0.516%), *Tenericutes* (from 0.100% to 0.242%), and *Bacteroidetes* (from 39.297% to 57.154%). At the genus level (Figure 5B), in comparison with normal mice, the abundances of *Escherichia-Shigella*, *the Lachnospiraceae NK4A136* group, and *Bacteroides* in the colitic model mice were 89.78, 2.19, and 1.93 times higher, respectively; meanwhile the abundances of *Alloprevotella*, *the Rikenellaceae RC9 gut group*, *Odoribacter*, *Alistipes*, and *Lactobacillus* in the model group were 43.66, 8.81, 3.81 3.68, and 3.57 times lower, respectively. Compared with the model group, the intervention with RT4 allowed a recovery in the abundance of *Escherichia-Shigella* (from 16.798% to 0.283%), *the Lachnospiraceae NK4A136 group* (from 6.123% to 6.729%), *Bacteroides* (from 24.050% to 11.600%), *Alloprevotella* (from 0.170% to 4.216%), *the Rikenellaceae RC9 gut group* (from 0.528% to 2.092%), *Odoribacter* (from 1.078% to 4.734%), *Alistipes* (from 0.898% to 2.857%), and *Lactobacillus* (from 5.662% to 21.263%). Subsequently, it was inferred that RT4 might play a role in regulating carbohydrate metabolism, amino acid metabolism, replication and repair, and membrane transport based on the analysis and the prediction provided by the fecal microbiota (Figure 5C). In summary, the conclusion was drawn that RT4 could restore the balance in the gut microbiota that had been induced by ulcerative colitis.

#### 2.1.7. Correlation between Gut Microbiota and UC Evaluating Indicators

In order to further analyze whether there is a correlation between gut microbiota at a genus level and the main UC evaluating indicators, the Spearman’s rank correlation statistical analysis method was used to evaluate the degree of relationship. The main indicators included body weight, DAI, colon length, spleen index, inflammatory mediators in serum and colon tissue, and the content of total SCFAs.

The results (Figure 5D) showed that four gut microbiota (*Alistipes*, *Odoribacter*, *Alloprevotella*, *the Rikenellaceae RC9 gut group*) were positively correlated with five indicators (SCFAs, IL-10, colon length, spleen index, body weight) and inversely related with four indicators (DAI, IL-6, TNF-*α*, IL-1*β*). It was also indicated that the other four gut microbiota *(the Lachnospiraceae NK4A136 group*, *Bacteroides*, *Escherichia-Shigella*, *Lactobacillus*) were positively correlated with four indicators (DAI, IL-6, TNF-*α*, IL-1*β*) and inversely related with five indicators (SCFAs, IL-10, colon length, body weight, spleen index). On the whole, six gut microbiota (*Alistipes*, *Odoribacter*, *Alloprevotella*, *the Rikenellaceae RC9 gut group*, *the Lachnospiraceae NK4A136 group*, *Escherichia-Shigella*), showed a significant correlation (*p* < 0.01, *p* < 0.05) with UC indicators, and it was posited that these might be the key microbiota contributing to the effect of RT4 for anti-UC.

### 2.2. Effect of RT4 on LPS-Induced Epithelial Barrier Damage Caco-2 Cells

#### 2.2.1. Cell Viability

The toxicity of RT4 to caco-2 cells was measured using a CCK-8 assay. The survival rates of caco-2 cells treated with 4, 8, and 16 μg/mL of RT4 were all greater than 80% (Figure 6A). However, the cell viabilities treated with 32 and 64 μg/mL of RT4 were 74.04% ± 4.65%, and 69.50% ± 5.44%, respectively. Therefore, three concentrations (4, 8, and 16 s) were selected for the doses of RT4 for subsequent studies.

#### 2.2.2. Anti-Inflammatory Activity

As shown in Figure 6C, the levels of three pro-inflammatory factors were apparently increased (*p* < 0.01) in the LPS-induced UC model group compared with the normal group. When compared with the model group, the levels of three pro-inflammatory factors were dose-dependently down-regulated in the RT4 treatment groups (*p* < 0.05, *p* < 0.01). In particular, RT4-M and RT4-H obviously down-regulated the levels of pro-inflammatory factors (*p* < 0.01). RT4-L apparently reduced the level of IL-6 (*p* < 0.05), and there were no obvious differences in the levels of IL-1*β* and TNF-*α* observed between the RT4-L group and the model group.

#### 2.2.3. Protection of Integrity of Caco-2 Monolayers

Since caco-2 cells could differentiate to form a epithelial cell monolayer and thus could simulate the absorption and barrier function of the small intestine [23], these cells are usually employed in the evaluation of the impact of drugs on the intestinal barrier. The alleviating effect of RT4 on LPS-induced intestinal epithelial barrier integrity destruction was monitored with transient electrical resistance (TEER) values (Figure 6B). It was shown that the resistivity of caco-2 intestinal epithelial cells decreased significantly after being stimulated with LPS; however, the TEER values of the RT4-H group were obviously higher than that of the model group, which suggested that RT4 could improve the barrier function of caco-2 cells.

#### 2.2.4. Effects of RT4 on the Expression of Tight Junction Proteins (TJPs)

Tight junctions are an integral component of the intestinal epithelial barrier, and the dysregulation of TJPs would lead to the defects in the epithelial barrier function [3]. Because the TJPs are expressed on the cell membrane, immunofluorescence technology could be used to intuitively observe the distribution of TJPs (such as ZO-1, occludin, claudin-1, and E-cadherin) there. The immuno-fluorescence results are shown in Figure 7. In the normal group, TJPs were continuously distributed with the strongest fluorescence intensity. Although 1 μL/mL of LPS did not inhibit the survival of caco-2 cells, it was able to reduce the expression of four TJPs, which led to the disordered distribution of ZO-1 or even to the disappearance of occludin, claudin-1, and E-cadherin. The intervention of three dosages of RT4 was able to dose-dependently increase the expression of four TJPs to varying degrees, and recover their distribution on the cell membrane. Based on these results, it could be judged that RT4 was able to alleviate the damage caused by LPS to the epithelial barrier in caco-2 cells.

### 2.3. Pharmacokinetic and Tissue Distribution Studies of Intragastric Dosing RT4 in DSS-Induced UC Mice

#### 2.3.1. Validation in the LC-MS Method

Specificity: Under the current experimental condition, the retention times of RT4 and PD (internal standard, IS) were 0.58 min and 3.68 min, respectively. Following a comparison of the chromatograms of blank biological samples, blank biological matrices spiked with RT4 samples, and medicated biological samples (2 h after intragastric administration of RT4 in UC mice) (Figure 8), it was clarified that there was no obvious endogenous substance interference from the biological matrix (plasma or tissues) within the retention times of both RT4 and PD.

Linearity and Lower Limit of Quantification (LLOQ): The regression equations, correlation coefficients (*R*^2^), linearity ranges, and the precision and accuracy of LLOQ are all listed in Table 1. In the range of 5–2500 ng/mL, the peak area depended linearly on the concentration of RT4 in plasma and the 12 tissue homogenates. The LLOQs in plasma and the 12 tissue homogenates were all 5 ng/mL with a signal-to-noise ratio over 10. The precision of LLOQ was 11.11~18.64%, and the accuracy of the LLOQ ranged from 89.63% to 116.96%. All the *R*^2^ values were greater than 0.9988. Therefore, the accuracies of LLOQ were all within ±20%, and the precisions were all less than 20%.

Precision and Accuracy: As shown in Table 2, the precision and accuracy of the methodology were analyzed with six replicates of QC samples (plasma, liver, or intestine tract homogenates) at three concentration levels, respectively. The precisions for RT4 in UC mice plasma, liver homogenates, or intestine tract homogenates were below 10.64%, 11.09%, and 9.38%, respectively. The accuracies were within the ranges of 95.62–108.51%, 94.47–102.58%, and 95.80–104.21%, respectively. The precisions were all less than 15%. The accuracies fell from the range of 85% to 115%.

Extraction Recovery and Matrix Effect: The summary of the extraction recoveries and matrix effects is presented in Table 3. The extraction recoveries of RT4 were in the range of 91.02 to 96.58%. The matrix effects of RT4 were between 93.03% and 97.67% indicating no obvious ion enhancement or suppression effect on the RT4 analytes. All the recoveries and matrix effects were within ±15%.

Stability: The stock solution stability analysis showed the mean% changes were all less than 2%. The plasma matrix stability and the autosampler stability analysis results are shown in Table 4. The mean% change in spiking RT4 to plasma under all stable conditions (short-term, long-term, three successive freeze–thaw cycles) was within the acceptable limits, namely the accuracy (deviation from the nominal concentration) and precision (*RSD*) were all within ±15%. All the samples were considered stable.

Effect of Dilution: The plasma samples with 7500 ng/mL of RT4 were diluted to one-third with blank UC mice plasma. The actual concentration was 2711.61 ± 107.36 ng/mL, with the precision being 6.58% and the accuracy being 108.46%. The results indicated that the method could accurately quantify the biological samples with concentrations exceeding the upper limit of quantification (ULOQ).

Carryover: After repeated injection of ULOQ samples, the peak areas in blank plasma samples were all less than 20% of the peak area of LLOQ concentration of RT4 and less than 5% of IS, indicating that there were no significant carryover effects in this method.

To sum up, the established LC-MS analytical method has the acceptable selectivity, a good linear range and stability, high precision and accuracy, reliable dilution, and a lower matrix effect and residue. The methodology could be used for the determination of RT4 content in biological samples.

#### 2.3.2. Pharmacokinetic and Tissue Distribution Study

Pharmacokinetics were further studied in greater depth to gain a fuller understanding of the actual absorbability of RT4 under UC pathological conditions. The method established based on UPLC-MS/MS was able to successfully determine the concentration of RT4 in the UC mice plasma. DAS 3.0 software was used to analyze the pharmacokinetic characteristics of RT4 in mouse plasma. The plasma concentration–time profiles of RT4 after oral administration and intravenous administration are shown in Figure 9. The main pharmacokinetic parameters analyzed with a compartment model are summarized in Table 5. After the oral administration of RT4 (10, 20, 40 mg/kg), the maximum plasma concentrations (*C*_max_:165.13 ± 46.75, 295.59 ± 74.25, 658.65 ± 264.01 µg/L) were achieved within 1~2 h (*T*_max_: 1.5 ± 1.22, 1.17 ± 0.41, 1.33 ± 0.52 h) post-dosing. The elimination half-life (*t*_1/2_) of RT4 in UC mice was found to be 4.67 ± 1.34, 6.29 ± 2.18, and 7.17 ± 2.78 h, respectively. The volumes of distribution (*V*_d_) were 61.91 ± 26.25, 71.24 ± 31.35, and 71.6 ± 24.78 L/kg, respectively. The clearance rates (*CL*) were 9.12 ± 2.12, 8.04 ± 2.95, and 7.2 ± 1.97 L/h/kg, respectively. Furthermore, the area under the curve (*AUC*_0-t_) were 991.55 ± 216.60, 2438.90 ± 568.77, and 5288.20 ± 1787.98 µg·h/L, respectively. The *AUC*_0-t_ in UC mice after intravenous administration of 2 mg/kg RT4 was 1246.37 ± 352.57 µg·h/L. By calculating the ratio of *AUC*_0-t_ in oral administration and intravenous administration, the absolute bioavailability of RT4 was 18.90 ± 2.70%. It is worth noting that the concentration–time curves showed a double peak phenomenon; the first peak time was about 1–2 h, suggesting a rapid capacity to absorb the compound into plasma in UC mice. The second peak time was about 4 h. Hepatointestinal circulation might be the reason for the fluctuations in blood concentration, or double absorption peaks in the oral PK curve. After RT4 was absorbed into the blood and entered the liver, the prototype drug was excreted into the gallbladder through bile, and then entered the duodenum with the secretion of bile. It was then re-absorbed into the blood, resulting in secondary absorption. This indicated that the oral administration of RT4 displayed both a high level of exposure and a slow elimination rate.

The distribution of RT4 in different tissue samples of UC mice at each time point after oral administration was explored by using the validated LC-MS method. The tissue distribution profiles are displayed in Figure 10. After the oral administration of 10, 20, and 40 mg/kg RT4 to UC mice, the compound was rapidly distributed in nearly all tissues, and the concentration trend in tissues was similar to that in plasma, reaching the highest concentration level at about 1–2 h. However, there were significant differences in the distribution of RT4 among different tissues. After 1–2 h of oral administration, RT4 was mainly distributed in the stomach, small intestine, and the colon. After 2–4 h, RT4 was mainly distributed in the small intestine and the colon, and the concentration of RT4 in the colon was higher than in the intestine. After 8 h, RT4 was mainly distributed in the colon. RT4 was scarcely detectable in the brains of UC mice. The high distributions in the colon, small intestine, and stomach (Figure 11) were consistent with the parameter *V*_d_ obtained in the pharmacokinetic study. As a result, it could be inferred that the colon was considered to be the main target organ for RT4 in alleviating UC.

## 3. Discussion

UC is often caused by a disturbance in gut microbiota and the destruction of colonic barrier function [1]. Meanwhile, the metabolites of intestinal microflora (such as SCFAs) may also play a significant role in affecting the integrity of the mucosal barrier [2]. There is a large network of cross-feeding interactions between the gut microbiome and the metabolites they produce [24,25]. The existing clinical medications available for UC are limited, alongside the core treatment indicators which are also limited. The clinical needs for UC treatment have not yet been met; for example, the mucosal healing rate remains less than 50% [26]. Any anti-inflammatory therapy requires long-term maintenance, which poses safety risks, and there is also a high risk of recurrence once the therapeutic intervention is withdrawn. The highly specific distribution of intestinal inflammation sites, as well as mucosal healing, are particularly important treatment strategies. RT4 (a natural product) showed positive anti-inflammatory effects in LPS-induced RAW 264.7 cells and in DSS-induced UC mice (intraperitoneal injected with RT4) [20]. However, oral administration is more suitable for treating ulcerative colitis. The immediate questions need to be further studied in depth. For example, the effects on the intestinal mucosal barrier, gut microbiota, UC-related indexes, pharmacokinetics, and the distribution of oral administrations of RT4 in mice are all unclear and remain to be clarified. For the first time, this study has explored the protective effect, the dynamic characteristics, and the tissue distribution of an oral administration of RT4 in UC mice. It was shown that the intragastric administration of RT4 to UC mice was not only able to alleviate the UC-related clinical symptoms, the inflammatory response, and SCFAs in a dose-dependent manner, but it could also improve the disorders in the gut microbiota and repair the damaged intestinal mucosal barrier in DSS-induced UC mice and in LPS-stimulated caco-2 cells (Figure 12). These results were confirmed through study of the intestinal flora, histopathological examination, transmission electron microscopy analysis, transient electrical resistance values, and tight-junction protein expression. This study innovatively discovered the treatment of the oral administration of RT4 for ulcerative colitis in mice. Based on the above, the superiority of oral administration over intraperitoneal injection can be elucidated as follows: First of all, oral administration is more convenient with no direct damage to the skin and mucosa, and patients displayed better rates of compliance with the drug regimen. Additionally, in terms of weight, DAI score, colon length, and inflammatory factors (TNF-*α*, IL-1*β*, IL-10), the effects of oral administration of RT4 are basically consistent, compared with the pharmacological effects of the intraperitoneal injection of RT4 reported in literature. Moreover, the administration of RT4 by gavage could selectively enrich it in the colon, which could target the treatment of ulcerative colitis. In particular, the impacts on body weight, DAI score, colon length or spleen index, IL-6 and IL-1*β* in the colon, SCFAs content, and histopathological examination of a high dosage of RT4 showed similar results to the effects of the standard treatment, SASP (*p* > 0.05).

The imbalance in the gut microbiota is one of the characteristics of UC. The destruction of intestinal microbiota accompanied by an increase in intestinal permeability leads to the destruction of intestinal epithelial function and triggers an immune response [2]. As for the gut microbiota study, on the one hand, the abundance of harmful bacteria such as *Escherichia-Shigella* was reduced via the intervention of RT4. On the other hand, RT4 treatment significantly enriched the abundance of beneficial bacteria including *Alloprevotella*, *the Rikenellaceae RC9 gut group*, *Alistipes*, *Lactobacillus*, *the Lachnospiraceae NK4A136 group*, and *Odoribacter.* Among these, the first three bacteria could produce SCFAs (having anti-inflammatory and antioxidant effects) in the intestine [27,28,29]. *Lactobacillus* is reported to be related to the protective effects of the intestinal barrier [30]. *Odoribacter* can reduce the inflammatory response and the probability of developing colon cancer [31]. In our study, the Spearman’s rank correlation between the intestinal microbiota and the analysis of the UC-evaluation indicator (body weight, DAI, colon length, spleen index, inflammatory mediators, total SCFAs content) indicated that six gut microbiota (*Escherichia-Shigella*, *Alloprevotella*, *the Rikenellaceae RC9 gut group*, *Alistipes*, *the Lachnospiraceae NK4A136 group*, *Odoribacter*) showed a significant correlation (*p* < 0.01, *p* < 0.05) with the above indicators of UC. It has been reported in the literature that facultative anaerobic bacteria were increased and the obligately anaerobic producers of SCFAs were decreased in UC patients [32]. In our study, there was an increase in facultative anaerobic bacteria (such as *Escherichia-Shigella*), and some obligately anaerobic producers (such as *Alistipes*, *Alloprevotella*, *the Rikenellaceae RC9 gut group*) related to SCFAs were relatively reduced in UC mice, which was consistent with the microbial community in UC patients. Furthermore, we also discovered that RT4 was able to increase the contents of SCFAs in mice cecal contents. Overall, both our research results and the results in the literature provide support to the conclusion that the improvement in gut microbiota composition contributed to the anti-inflammatory and intestinal barrier protective effects of RT4 for anti-UC treatment.

As for the intestinal mucosal barrier study, in UC mice, RT4 could restore the integrity of intestinal mucosal cells and the orderly arrangement of microvilli [33]. In particular, almost no degeneration in organelles in the RT4-H group was observed, and the ultrastructure of the colonic mucosal epithelial cells was well maintained. In LPS-stimulated caco-2 cells, the TEER values of the RT4-H group were increased, and the expression and the distribution of TJPs (such as ZO-1, occludin, claudin-1, and E-cadherin) could either be increased or recovered. On the basis of the above results, it can be concluded that RT4 was able to alleviate the damage caused by LPS to the intestinal barrier of caco-2 cells or UC mice.

Patients with UC have many clinical manifestations. Hemorrhagic diarrhea is the most common early symptom. Other symptoms include abdominal pain, bloody stools, weight loss, urgency and heaviness, and vomiting, etc. The common diagnostic tests include blood samples and colonoscopies. The etiology and pathogenesis of UC are related to the interaction of multiple factors such as environmental factors, genetic factors, and gut microbiota, leading to an intestinal immune imbalance (imbalance of inflammatory/anti-inflammatory factors). In the current study, the therapeutic effects of orally administered RT4 on UC in Balb/c mice induced by DSS were investigated. Key findings included the reduction in DAI, the restoration of colon length, decrease in pro-inflammatory cytokines, an increase in anti-inflammatory cytokine levels, improvement in the colonic histopathology and ultrastructure, modulation of the composition of gut microbiota, and an increase in the content of SCFAs. Except for colon length, the DAI, inflammatory/anti-inflammatory factors, colonic histopathology, gut microbiota, and the content of SCFAs all have relevance and translatability to human UC.

Overall, the primary reason for RT4’s effect on UC is as follows: under the intervention of RT4, the gut microbiota were regulated, leading to an increase in the content of metabolite SCFAs (serving as energy sources for colonic cells and anti-inflammatory agents), a decrease in the secretion of inflammatory factors caused by gut microbiota, a relief of the imbalance between inflammatory and anti-inflammatory factors, a reduction in sustained inflammation of the intestinal mucosa, and a reduction in intestinal mucosal barrier damage.

In the assessment of the absorption and tissue distribution of the oral administration of RT4 in UC mice, RT4 exhibited a moderate oral availability with the absolute bioavailability being 18.90 ± 2.70%. RT4 could be rapidly absorbed with the *T*_max_ being 1~2 h (1.5 ± 1.22 h, 1.17 ± 0.41 h, 1.33 ± 0.52 h), and it could be slowly eliminated with the *t*_1/2_ being 4.67 ± 1.34 h, 6.29 ± 2.18 h, and 7.17 ± 2.78 h, respectively, which indicated that RT4 could be absorbed into the blood after oral administration to UC mice, the maximum plasma concentration could be reached in 1~1.5 h, and the plasma concentration could be maintained at a certain level within 5~7 h. In general, compared with the current clinical drugs for UC, RT4 has the characteristics of rapid absorption and slow elimination, and it could be specifically distributed in colonic tissue. This is closely related to the therapeutic effect of RT4 on UC. In addition, RT4 was specifically distributed in the intestinal tissue, which provided an explanation for the specific treatment of UC with RT4.

As we all know, ulcerative colitis is a chronic inflammatory disease, and the models established in the current research tend to be of acute inflammation. Therefore, in future research, we will continue to establish chronic inflammation models for the further investigation of the role of RT4 in ulcerative colitis. In further studies, an evaluation of its safety and toxicity should be a priority. Additionally, both cell models and mice models should be utilized to corroborate the effect of specific microbiota on inflammation.

## 4. Materials and Methods

### 4.1. Materials and Animals

RT4 and Panaxadiol (PD, CAS: 19666-76-3) were prepared by us, with the purity of 98.5% and 99.1% (assayed by the ELSD-HPLC method), respectively. SASP (lot: RH143735) and DSS (lot: C14608113, MW: 40,000 Da) were purchased from Shanghai Baishun Biotechnology Co., Ltd. (Shanghai, China), and used as the positive control drug and model-inducing drug, respectively. LPS was sourced from the Sigma-Aldrich Company (Shanghai, China).

Enzyme-linked immunosorbent (ELISA) kits for the determination of TNF-*α*, IL-6, IL-1*β*, and IL-10 in mouse were purchased from MultiSciences (Lianke) Biotech Co., Ltd. (Hangzhou, China). MEM medium and fetal bovine serum were purchased from Thermo Fisher Scientific (Waltham, MA, USA). The Mouse SCFA ELISA kit was provided by Shanghai Zhen Ke Biological Technology Co., Ltd. (Shanghai, China). ZO-1 (21773-1-AP), claudin-1 (28674-1-AP), occludin (66738-1-Ig), E-cadherin (60335-1-Ig), TPH (DF6465), and *β*-actin (81115-1-RR) antibody were all purchased from Proteintech Group, Inc. (Wuhan, China). Other chemicals and reagents used in current study were of analytical pure.

Male Balb/c mice (6–7-week-old) were obtained from YISI Experimental Animal Technology Co., Ltd. (Changchun, China). Complying with the “Guide for the Care and Use of Laboratory Animals”, the mice were raised in an environment with the relative humidity being 60% ± 5% and room temperature being 25 ± 2 °C, and the light/dark cycle of 12 h.

### 4.2. Effect on DSS-Induced Mice

#### 4.2.1. Effect on DSS-Induced Mice

The mice adaptively raised for one week were randomly divided into 6 groups (*n* = 10): normal control group, model group (DSS), positive drug group (DSS + 200 mg/kg of SASP), RT4-L group (DSS + 10 mg/kg of RT4), RT4-M group (DSS + 20 mg/kg of RT4), and RT4-H group (DSS + 40 mg/kg of RT4). The normal group drank normal water ad libitum, while the other groups drank 3% DSS (*w*/*v*) aqueous solution ad libitum for 7 consecutive days to induce the UC model.

From Day 8 to Day 10, mice in each group drank water normally. The mice in the normal control group and model group received a daily intragastric dose of 0.5% CMC-Na solution, whereas the other groups received SASP or RT4-CMC-Na solutions daily via intragastric administration. The intragastrically administrated volumes were all 10 mL/kg.

#### 4.2.2. Body Weight and DAI

Throughout the experiment, mice were weighed daily before administration, and the fecal consistency and fecal occult blood data were also recorded. The DAI score—a composite score based on weight loss, fecal features, and scores of blood stool—was calculated to assess and quantify the UC symptoms [21]. The DAI score represents one-third of the total of the above three scores.

#### 4.2.3. Sample Collection and Pretreatment

On the 11th day, after being fasted for 12 h, the fresh mouse feces of the normal group, model group, and RT4-H group were collected and quickly frozen at −80 °C for gut microbiota analysis.

Mice were anaesthetized with 0.3% pentobarbital sodium (10 mL/kg, i.p.). The whole blood was drawn from the abdominal aorta, and then was centrifuged (3000 rpm) for 15 min to obtain the serum; it was then stored at −80 °C for the following determination of inflammatory factors. Then, colon tissues were quickly taken and were measured for their length. The colon was next rinsed with precooling physiological saline and dried with filter paper. A part of the colonic tissue was used for histopathological examination and was placed in a 4% paraformaldehyde solution, while the other part was used for electron microscopy determination and was placed in 4% glutaraldehyde solution. In addition, the remainder of the colonic tissues was weighed and homogenized in nine-fold (*w*/*v*) cold PBS, which was then centrifuged (10,000 rpm) for 5 min at 4 °C to acquire the supernatant that was then used to detect the inflammatory cytokines.

The spleen was collected for spleen index analysis. The calculation formula for the spleen coefficient is as follows:Spleen index = Spleen mass (mg)/Mice body weight (g)

The fresh cecal contents were collected for SCFAs analysis.

#### 4.2.4. Inflammatory Factor Determination

According to the instructions of the ELISA kits’ manufacturer, the absorbance at 450 nm was measured on a microplate reader, and the serum levels or colonic levels of inflammatory factors (TNF-*α*, IL-6, IL-1*β*, and IL-10) were calculated.

#### 4.2.5. Histological Examination

The colonic tissue was fixed with 4% paraformaldehyde fixative, dehydrated in gradient ethanol, embedded in paraffin, sectioned, stained with hematoxylin and eosin, and finally observed under a microscope.

#### 4.2.6. Transmission Electron Microscopy (TEM) Examination

The colonic tissues fixed in 2.5% glutaraldehyde were sectioned, post fixation was performed, then the tissues were dehydrated using a gradient, resin-embedded, cured, cut into thin sections, and stained with uranyl acetate and lead citrate. The observations were performed using transmission electron microscopy.

#### 4.2.7. Determination of Total SCFAs in Cecal Samples

The fresh cecal content (about 0.3 g) was accurately weighed, and then homogenized with PBS (9 mL/g) on ice, followed by centrifuging (3000 rpm) for 10 min at 4 °C. A mouse SCFA enzyme-linked immunosorbent assay kit was used to determine the contents of total SCFAs in supernatants. The absorbance was measured using a microplate reader at 450 nm, and the contents were calculated.

#### 4.2.8. Gut Microbiota

The 16s rDNA test was performed and analyzed by Shanghai Applied Protein Technology Co., Ltd. (Shanghai, China).

The total DNA of fresh fecal samples in UC mice was extracted using the CTAB/SDS method. Then, the quantity and purity of the DNA were evaluated on 1% agarose gels. In order to sequence the structure of the gut microbiota, 341F-806R- and 1380F-1510R-specific barcoded primers were used to amplify the V3–V4 region of the 16S rRNA and V9 region of the 18S rRNA of each sample, respectively. All PCR reactions were carried out with the Phusion^®^High-Fidelity PCR Master Mix (New England Biolabs (Beijing, China)). The PCR products were added with SYB green loading buffer, and electrophoresis detection was performed with a 2% agarose gel. Samples with bright main strips between 400 and 450 bp were selected for further experiments. The mixed PCR products were purified using an AxyPrepDNA Gel Extraction Kit (AXYGEN). The NEB Next^®^Ultra™DNA Library Prep Kit for Illumina (Beijing, China) was used to sequence libraries and add index codes. Library quality was evaluated on the Qubit@ 2.0 Fluorometer (Thermo Scientific, Waltham, MA, USA) and Bioanalyzer 2100 System (Agilent (Santa Clara, CA, USA)). Then, sequencing was performed on an Illumina NovaSeq 600 platform, and 250 bp paired-end reads were generated.

The UPARSE-OTU and UPARSE-OTUref algorithms in the UPARSE software (version 7.0.1001) package were used to analyze the paired-end reads. Alpha and beta diversities were both analyzed using in-house Perl scripts. Sequences with ≥97% similarity were considered as the same OTUs, and a representative sequence was then selected to annotate taxonomic information by the RDP classifier. The OTU table was rarified to compute the alpha diversity based on calculating the Shannon index metrics. A Krona chart was used to visualize the relative abundance of bacterial diversity. Prior to cluster analysis, principal component analysis (PCA) was performed to reduce the dimensionality of the original variables. QIIME software (version 1.9.1) was used to calculate the weighted unifrac distance. Principal coordinate analysis (PCoA) was then applied to obtain principal coordinates and visualization.

### 4.3. Effect on LPS-Induced Epithelial Barrier Damage Caco-2 Cells

#### 4.3.1. Cell Culture and Viability Assay

Caco-2 cells (Procell Life Science & Technology Co., Ltd., Wuhan, China) were maintained in the MEM medium supplemented with 10% heat-inactivated fetal bovine serum, 1% penicillin, and streptomycin at 37 °C and 5% CO_2_.

The cell viability assay was performed in 96-well plates. The caco-2 cells were seeded at a density of 1 × 10^4^ cells per well and cultured for 24 h at 37 °C and 5% CO_2_. Then, the cultures were treated with 0, 4, 8, 16, 32, and 64 μg/mL of RT4 and 10 μL of CCK-8 solution. After incubation for 4 h, the absorbance was measured at 450 nm, and the effect on the viability of caco-2 cells was calculated.

#### 4.3.2. Effect on TEER

Caco-2 cells (1 × 10^5^) were cultivated at 37 °C and 5% CO_2_ in the upper chamber of a transwell chamber with a 0.4 µm pore size. A blank hole without inoculating cells was set in the board, and the blank resistance value was also measured. A Millcell ERS-2 Transmembrane Resistance Meter (Millipore, Billerica, MA, USA) was used to test the resistances of monolayers which were assessed with TEER changes in caco-2 cells. TEER values were calculated based on the following formula:TEER (Ω·cm^2^) = (*R*_1_ − *R*_0_) × *A*

(*R*_1_: resistance value of the caco-2 cell compartment, Ω; *R*_0_: resistance value of the unseeded cell compartment, Ω; *A*: membrane area of the compartment, cm^2^).

The intestinal barrier model was considered as successfully established when the TEER values were greater than 400 Ω·cm^2^. Then, the cells were stimulated with LPS (1 μg/mL) for 24 h at 37 °C and 5% CO_2_ [22], followed with being incubated with RT4 (0, 4, 8, and 16 μg/mL) for 18 h.

#### 4.3.3. Immunofluorescence Cytochemistry

Caco-2 cells (4 × 10^5^) were inoculated onto the cell creep plate, allowing cells to adhere and achieve approximately 80% confluence. The following modeling method was the same as in Section 4.3.2, and the cells were also treated with RT4 (0, 4, 8, 16 μg/mL) for 18 h. After that, they were fixed in 4% paraformaldehyde for 20 min at room temperature, washed with precooling PBS for 5 min three times, and then treated with 0.1% Triton X-100 for 10 min at room temperature. Subsequently, caco-2 cells were blocked in PBS-containing BSA (2%) for 1 h at room temperature. Afterward, these cells were incubated with occludin, claudin-1, ZO-1, and E-cadherin primary antibodies for 12 h at 4 °C. After washing cells with cold PBS three times, FITC and Cy3-labeled goat anti-rabbit IgG secondary antibodies were cultured in darkness at an ambient temperature for 1 h. DAPI was used to counter-stain the cover slips, then they were imaged with a fluorescence microscope (Olympus BX53 (Tokyo, Japan)).

### 4.4. Pharmacokinetic and Tissue Distribution Studies of Intragastric Administration RT4 in Mice with UC

#### 4.4.1. Chromatographic and Mass Spectrometry Conditions

The UPLC-MS/MS analysis was performed on an Acquity^TM^ UPLC and XEVO TQ-S Triple Quadrupole Tandem Mass Spectrometry (Waters Corp, Milford, MA, USA) equipped with an electrospray ionization (ESI) source system.

The chromatographic separation of RT4 and IS was achieved on an Acquity UPLC BEH C_18_ column (2.1 × 50 mm, 1.7 μm, Waters Corporation (Milford, MA, USA)) with the column temperature maintained at 30 °C. LC was performed at a flow rate of 0.5 mL/min using the gradient elution procedure (water as phase A and acetonitrile as phase B; 0–2 min, 50%→100% B; 2–4 min, 100% B; 4–4.1 min, 100%→50% B; 4.1–6 min, 50% B). The injection volume was 5 μL using an autosampler which was maintained at 4 °C.

For mass spectrometry detection, RT4 and IS were determined in positive ionization mode. The parameters were set as follows: The ion source temperature was 150 °C, the desolvation temperature was 350 °C, the capillary voltage was 3.0 kV, the desolvation gas flow was 700 L/h, and the cone gas flow was 150 L/h. MRM mode was used for quantification. RT4 was monitored under the protonation precursor ion [M + H]^+^, generated at *m*/*z* 655.4→*m*/*z* 143.5 with a cone voltage of 40 V and collision energy of 14 V. IS was monitored at the protonated precursor ion [M + H]^+^ to the product ion at *m*/*z* 461.1→*m*/*z* 425.1 with the cone voltage being 22 V and collision energy being 16 V. The daughter spectrum of RT4 and IS are shown in Figure 13. The data acquisition and analysis were performed using Masslynx software (version 4.1).

#### 4.4.2. Preparation of Standard Solutions and Quality Control Samples

RT4 and PD were accurately weighed and then dissolved in acetonitrile to obtain 100 µg/mL of stock solutions, respectively. Next, the desired working solutions (50, 250, 500, 2500, 5000, and 25,000 ng/mL) were prepared with stepwise dilution. Furthermore, the standard solution of IS, used for plasma or tissues (liver and intestine tract) homogenates spiking purposes, was diluted to 100 ng/mL. Then, 10% of spiking percentage in plasma or tissue homogenate volume was chosen, and six calibration standards (5, 25, 50, 250, 500, and 2500 ng/mL) solutions were accordingly obtained.

Low (LQC), middle (MQC), and high (HQC) quality control samples of 10, 200, and 2000 ng/mL were prepared, respectively.

#### 4.4.3. Sample Collection and Preparation

The mice with UC were divided into five groups: UC model group (intragastric administration, 0 mg/kg, *n* = 42), RT4-ig-L group (intragastric administration, 10 mg/kg, *n* = 42), RT4-ig-M group (intragastric administration, 20 mg/kg, *n* = 42), RT4-ig-H group (intragastric administration, 40 mg/kg, *n* = 42), RT4-iv group (intravenous administration, 2 mg/kg, *n* = 42). After administration once, orbital venous blood and tissue samples (heart, liver, spleen, lung, kidney, stomach, small intestine, colon, testis, muscle, fat, brain) were collected, respectively. For the intragastric administration groups, the sampling time points were at 0.5, 1, 1.5, 3, 6, 12, and 24 h with 6 mice per point. For the intravenous infusion administration group, the sampling time points were at 0.033, 0.167, 0.5, 1.5, 6, 12, and 24 h with 6 mice per point. Mice in the model group were taken and collected; the same samples were collected as above for methodology validation.

Centrifuge tubes containing heparin sodium were used for placing blood in a warm-bath at 37 °C for 30 min. Then, the tubes were centrifuged at 3000 rpm for 15 min to extract the upper layer of plasma. Each plasma sample (100 μL) was mixed with 400 µL IS (PD, 100 ng/mL) solution for deproteinization. The mixtures were vortexed for 1 min and centrifuged (10,000 rpm) for 15 min at 4 °C to obtain the supernatants, which were then freeze-dried to obtain the residues. After that, 100 µL acetonitrile was added into the residue to obtained the plasma test solution for UPLC-MS/MS analysis.

Each tissue was rinsed with pre-cooled physiological saline and dried with filter paper. Then, they were accurately weighed and homogenized in saline (4 mL/g tissue) on ice to obtain the homogenate. Each 400 μL of tissue homogenate was mixed with 800 μL IS solution, and then was treated in the same way as the plasma sample. The freeze-dried residue was dissolved in 100 μL acetonitrile to acquire the tissue test solution for UPLC-MS/MS analysis.

#### 4.4.4. Method Validation

The bioanalytical method validation was carried out according to the Chinese Pharmacopoeia (2020 version) guidance.

Specificity: The specificity was defined as noninterference at a retention time of RT4 or IS from the endogenous plasma or tissue homogenates. It was assessed by comparing the MRM chromatograms of blank plasma or tissue homogenates from six mice, blank plasma or tissue homogenates spiked with LLOQ concentrations of RT4 and IS (PD), and plasma or tissue homogenate samples obtained from a UC mouse 2 h after intragastric administration of RT4.

Linearity and LLOQ: The standard curves were constructed with the ratio of the measured area of RT4 to the area of IS as the *x*-axis and the concentration of RT4 as the *y*-axis, via analyzing calibration standards samples over three different days. The linear regression equation was calculated by weighted linear least squares regression (*w* = 1/*x*^2^).

LLOQ is typically defined as the lowest concentration in the linear range that could be quantitatively detected, with a signal-to-noise ratio of not less than 10 and its accuracy and precision should be with a relative standard deviation (*RSD*) < 20% and deviations from the nominal concentration ((measured value/true value) × 100%) within ±20% by six replicate analyses.

Precision and Accuracy: Six replicates of QC samples of high, medium, and low concentration were taken to assess the intra-day and inter-day precision and accuracy on three consecutive analytical days. QC samples were prepared once a day. The precision calculation should be based on the *RSD* of the concentration determined in all replicates which should be less than 15%. The accuracy of the method was calculated as the % nominal concentration and a range of 85% to 115% was required.

Extraction Recovery and Matrix Effect: The extraction recovery and matrix effect were assessed at three QC concentration levels with six duplicates for each concentration. The recovery and matrix effect should be within ±15%. The extraction recovery of RT4 was obtained through comparison of the mean peak area of RT4 in the extracted QC sample with the mean peak area of spiked post-extraction (extracted blank plasma or tissue spiked with standard solution) samples at the same levels. Matrix effects were determined by the ratio of the peak area of RT4 in the spiked post-extracted samples to the peak area of RT4 directly acquired from the pure mobile phase solution.

Stability: RT4 and IS stock solutions were kept at room temperature for 8 h, or at 4 °C for 4 weeks to perform the stock solution stability analysis. The RT4 stock solution was diluted to the middle QC level and it was compared it with 5 freshly configured duplicate samples to estimate the RT4 stability. The mean% change in RT4 and PD should be within ±10%. During sample processing, assessments of low and high levels of plasma-spiked QC samples with six repetitions considering different expected conditions were performed for the matrix stability analysis. The stabilities included the short-term stability (storage at 25 ± 2 °C for 24 h), the long-term stability (storage at −80 °C for 1 month), the freeze–thaw stability after three successive freeze–thaw cycles (frozen at −80 °C and subsequently thawed at 25 °C as one cycle), and the autosampler stability (storage in autosampler at 4 °C for 12 h). The stability was evaluated by comparing the measured concentration with the labeled concentration based on the standard curve obtained using freshly prepared calibration standards. The sample was considered stable with the deviation between the mean of each concentration and the labeled concentration within ±15%.

Effect of Dilution: The dilution effect was assessed with the dilution of 7500 ng/mL of RT4 to one-third with blank mice plasma to obtain theoretical concentrations of RT4 of 2500 ng/mL with six duplicates. The actual concentration was calculated based on the daily standard curve. Precision and accuracy should be within ±15%.

Carryover: The carryover was performed by extracting the blank plasma which was injected immediately after repeated injections of the ULOQ samples three times. The peak area in blank samples after ULOQ injection must be no more than 20% of the peak area of the LLOQ concentration of RT4 and no more than 5% of IS. The verification was performed continuously in triplicate.

#### 4.4.5. Statistical Analysis

The Masslynx 4.1 software (Waters (Louis, MO, USA)) was used to collect chromatograms and calculate peak areas. The pharmacokinetic parameters were calculated by DAS software (version 3.2.8, Chinese Pharmacological Association, Suzhou, China) using a non-compartment model. The statistical analyses of the data were performed by Dunnett’s *t*-test using SPSS 17.0 (IBM Company (Amonk, NY, USA)), and *p* < 0.05 was considered statistically significant. The figures were plotted with Microsoft Excel 2016 (Microsoft version 16.0 (Redmond, WA, USA)).

## 5. Conclusions

In the current study, DSS-induced ulcerative colitis mice and LPS-induced epithelial barrier damage caco-2 cells were used to investigate the effect of the oral administration of RT4 on UC. It was demonstrated that RT4 exerted an inhibitory effect on UC responses, which was manifested by the reductions in the DAI score, spleen index, and cytokine contents, coupled with increases in body weight, SCFAs, and extension of the colon length. In addition, RT4 could enhance the intestinal epithelial barrier function, which was evidenced by the reduction in intestinal damage, the maintenance of the ultra-structure of colonic mucosal cells, the increased TEER values, and the high tight-junction protein expression. Then, six key gut microbiota showing a strong correlation with UC indicators were identified as playing crucial roles in the therapeutic effects of RT4 on UC. Finally, the oral administration of RT4 to UC mice had the characteristics of fast absorption, slow elimination, and targeted distribution in the intestinal tissues. Overall, the above results indicated that RT4 could prevent the UC process by inhibiting inflammatory responses, and regulating the intestinal barrier and the balance of the gut microbiota. RT4 could serve as a potential candidate drug for the treatment of UC.

## Figures and Tables

**Figure 1 ijms-25-00835-f001:**
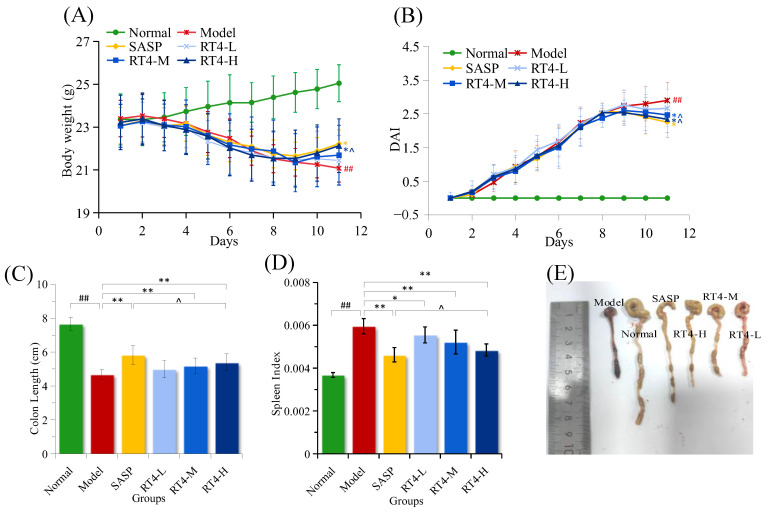
Effects of RT4 on DSS-induced colitis symptoms in mice. (**A**) Body weight; (**B**) Disease activity index (DAI); (**C**) Colon length; (**D**) Spleen index; (**E**) Representative photographs of the colon. (Compared with the normal group, ## *p* < 0.01; compared with the model group, ** *p* < 0.01, * *p* < 0.05; compared with the SASP group, ^ *p* > 0.05, Dunne’s *t*-test).

**Figure 2 ijms-25-00835-f002:**
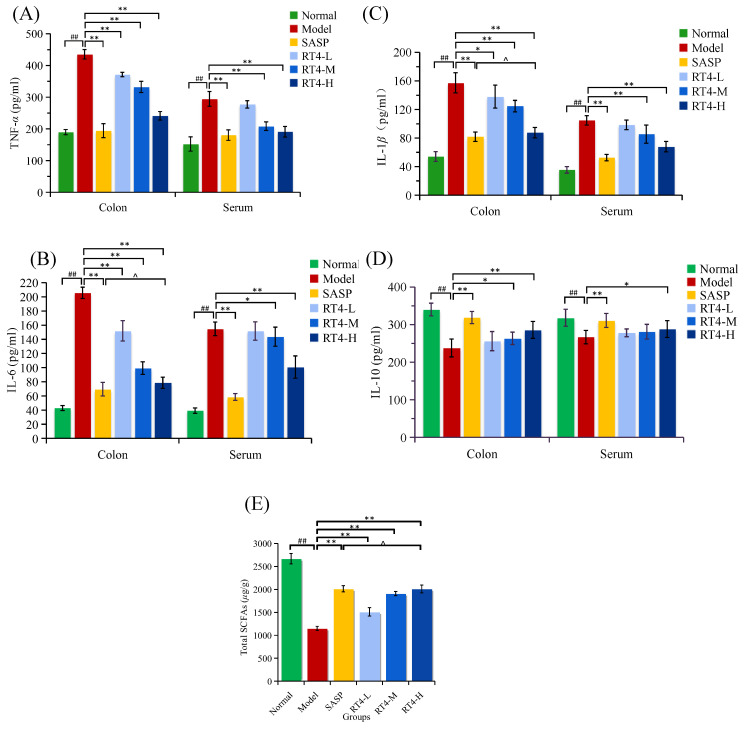
Effects of alleviating inflammation and increasing the content of total SCFAs: (**A**–**D**) Levels of TNF-*α*, IL-6, IL-1*β*, and IL-10 in the colon or in serum of mice; (**E**) Total SCFAs in the UC mice cecal contents. (## *p* < 0.01 vs. the normal group; ** *p* < 0.01, * *p* < 0.05 vs. the model group; ^ *p* > 0.05 vs. the SASP group, Dunne’s *t*-test).

**Figure 3 ijms-25-00835-f003:**
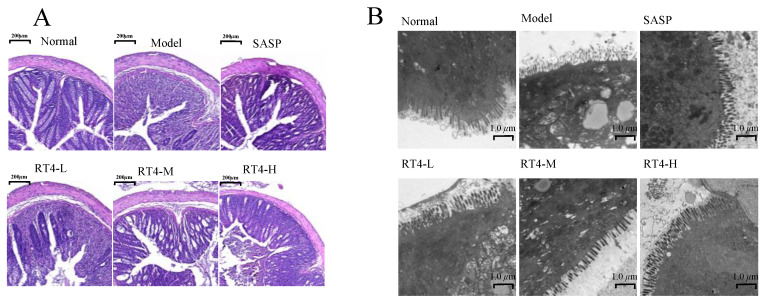
Effect of alleviating colonic mucosal injury: (**A**) H&E staining photos of the colon; (**B**) Transmission electron microscopy photos of the colon.

**Figure 4 ijms-25-00835-f004:**
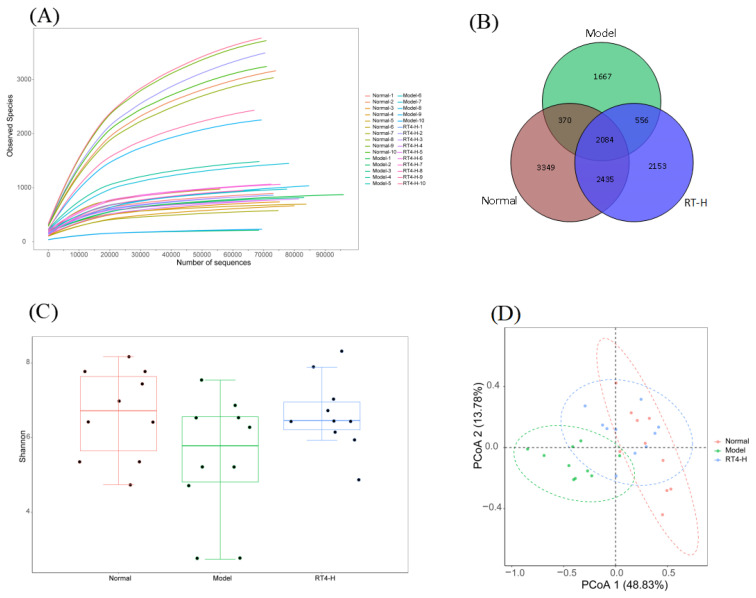
Effects of RT4 on the diversity and structure of the intestinal flora in the DSS-induced UC mice: (**A**) Rarefaction curves based on the OTU levels of the intestinal flora; (**B**) Venn diagram showing the common and unique OTUs between groups; (**C**) Shannon index analysis; (**D**) PCoA of the intestinal flora based on the weighted UniFrac distance.

**Figure 5 ijms-25-00835-f005:**
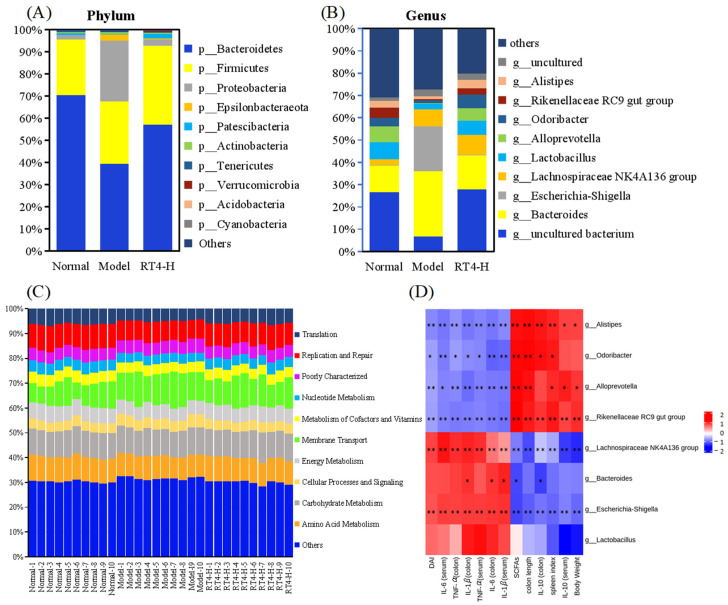
Effects of RT4 on the relative abundance of specific intestinal bacteria in the DSS-induced UC mice and the correlation UC evaluating indicators: (**A**) Relative abundance of the intestinal flora in each group at the phylum level; (**B**) Relative abundance of the intestinal flora in each group at the genus level; (**C**) Predicted function of differential microbiota; (**D**) Correlation heatmap based on Spearman’s rank correlation analysis, the red color represents a positive correlation, and the blue color represents a negative correlation. (** *p* < 0.01, * *p* < 0.05).

**Figure 6 ijms-25-00835-f006:**
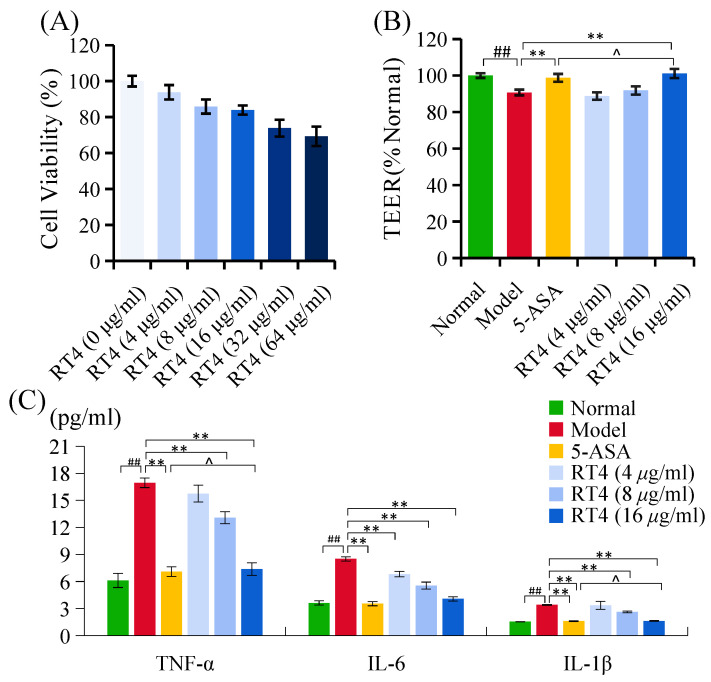
Effects of RT4 on the caco-2 cells: (**A**) Cell viability; (**B**) TEER; (**C**) Levels of TNF-*α*, IL-6, and IL-1*β*. (compared with normal group, ## *p* < 0.01; compared with the model group, ** *p* < 0.01, compared with the SASP group, ^ *p* > 0.05).

**Figure 7 ijms-25-00835-f007:**
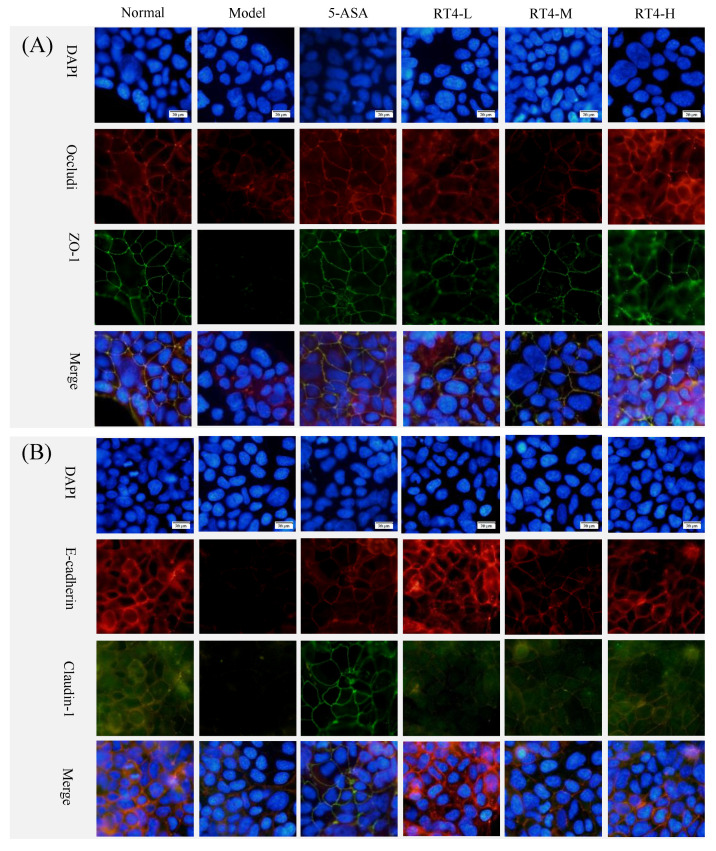
Effects of RT4 on tight-junction proteins: (**A**) ZO-1, occludin; (**B**) claudin-1, E-cadherin.

**Figure 8 ijms-25-00835-f008:**
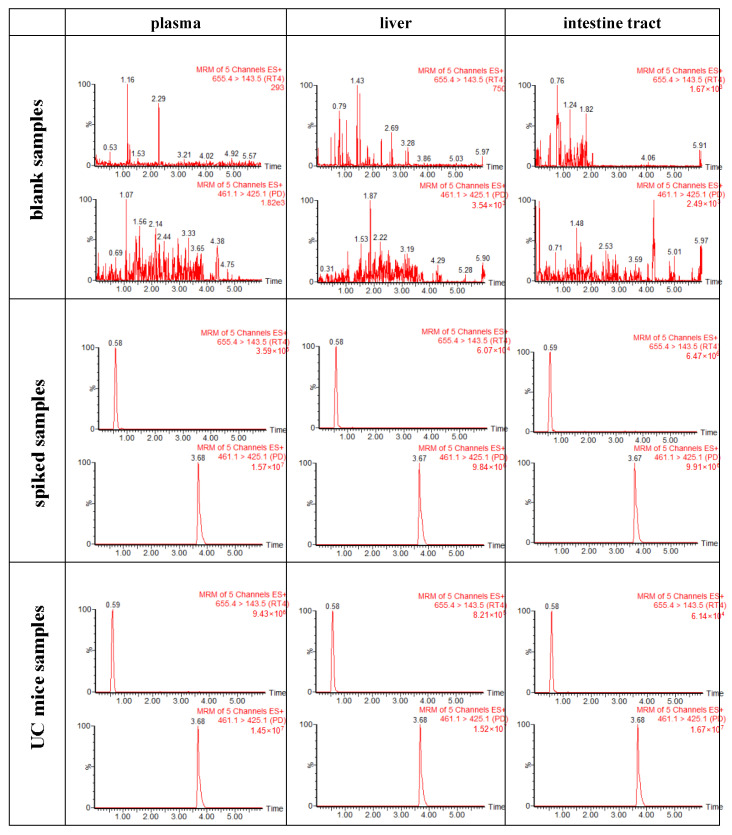
MRM chromatograms of RT4 (RT: 0.58 min; *m*/*z*: 655.4→143.5) and PD (RT: 3.68 min; *m*/*z*: 461.1→425.1).

**Figure 9 ijms-25-00835-f009:**
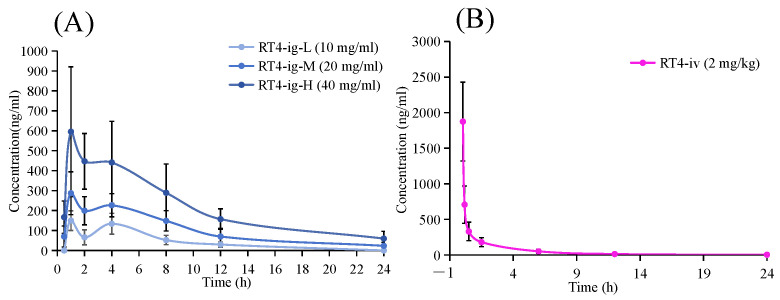
Mean plasma concentration-time curves of RT4 in the UC mice: (**A**) Curves receiving oral doses of 10, 20, and 40 mg/kg; (**B**) Curve receiving an intravenous dose of 2 mg/kg.

**Figure 10 ijms-25-00835-f010:**
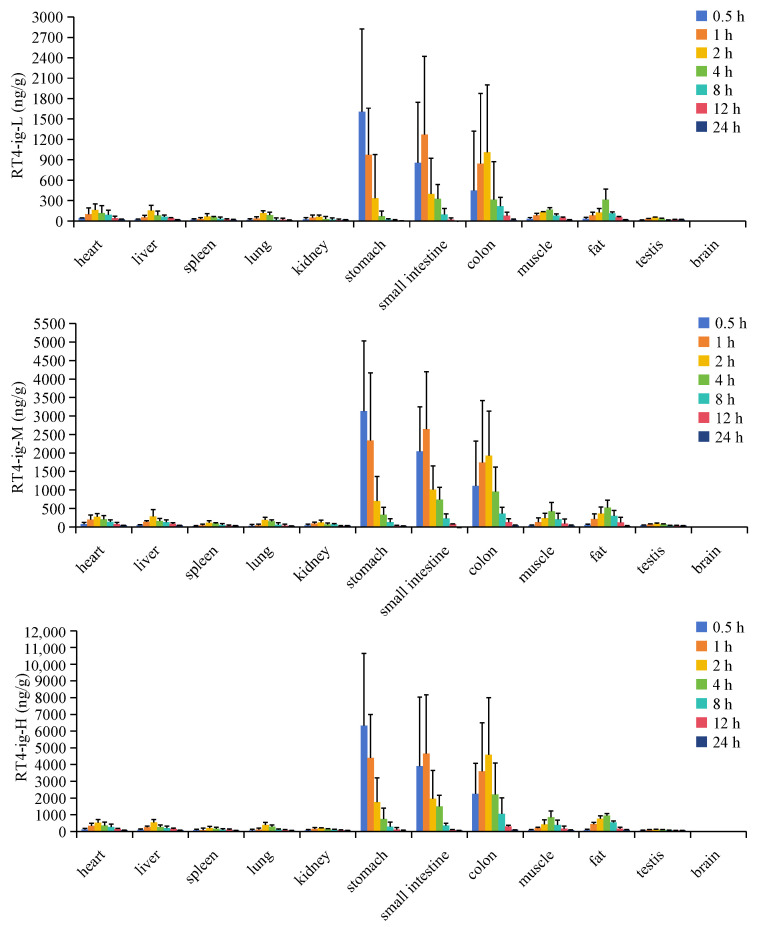
Tissue distribution of RT4 in the UC mice after being orally administrated.

**Figure 11 ijms-25-00835-f011:**
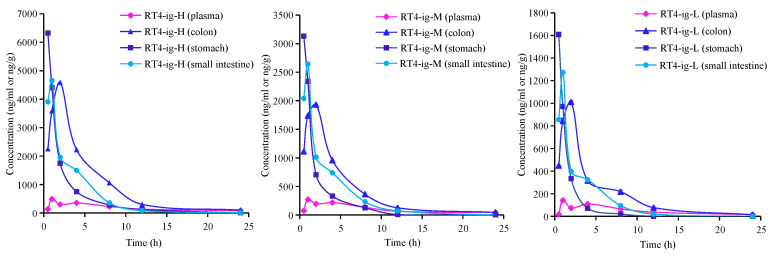
The mean plasma, stomach, small intestine, and colon distribution of RT4 in mice after oral administration of RT4 at different time points.

**Figure 12 ijms-25-00835-f012:**
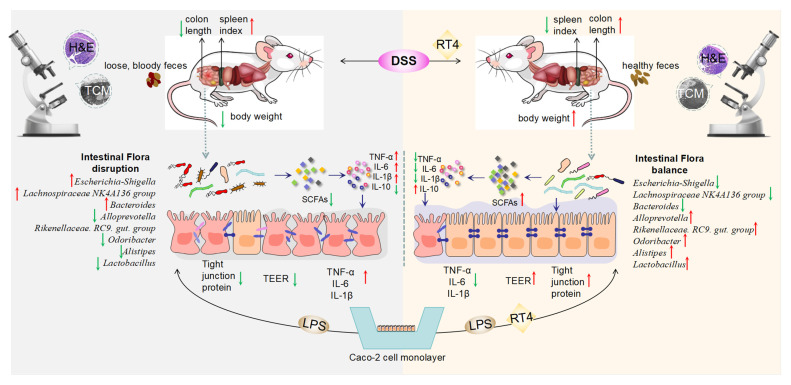
The effects of RT4 on ulcerative colitis (The red arrow represents an increase, while the green arrow represents a decrease).

**Figure 13 ijms-25-00835-f013:**
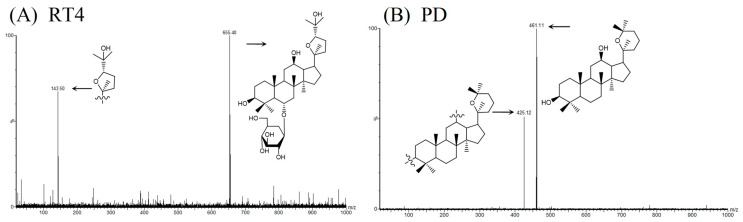
Mass spectra and the proposed fragmentation patterns: (**A**) RT4; (**B**) PD (IS).

**Table 1 ijms-25-00835-t001:** Calibration curves, correlation coefficients, linear range, precision, and accuracy LLOQ of RT4 in the UC mice samples.

Sample	Calibration Curve	Correlation Coefficient (*r*^2^)	Linear Range (ng/mL)	LLOQ (%)	
				Precision	Accuracy
plasma	*y* = 0.0005 *x* + 0.0037	0.9992	5–2500	13.72	92.01
heart	*y* = 0.0004 *x* − 0.0024	0.9994	5–2500	13.36	92.25
liver	*y* = 0.0005 *x* + 0.0023	0.9992	5–2500	17.70	91.36
spleen	*y* = 0.0004 *x* − 0.0082	0.9988	5–2500	13.50	113.45
lung	*y* = 0.0004 *x* − 0.0009	0.9999	5–2500	12.65	106.58
kidney	*y* = 0.0004 *x* − 0.0030	0.9996	5–2500	11.11	89.63
stomach	*y* = 0.0005 *x* + 0.0090	0.9986	5–2500	17.20	106.24
Intestine tract	*y* = 0.0004 *x* + 0.0073	0.9993	5–2500	14.96	108.82
colon	*y* = 0.0004 *x* + 0.0018	0.9998	5–2500	18.64	115.90
muscle	*y* = 0.0004*x* − 0.0072	0.9994	5–2500	18.71	92.01
fat	*y* = 0.0004 *x −* 0.0087	0.9991	5–2500	11.95	116.96
testis	*y* = 0.0004 *x −* 0.0050	0.9996	5–2500	16.01	111.22
brain	*y* = 0.0005*x* − 0.0088	0.9992	5–2500	18.44	113.83

**Table 2 ijms-25-00835-t002:** Precision and accuracy of RT4 in plasma, liver, and intestinal tract of UC mice.

Sample	Concentration (ng/mL)	Intra-Day			Inter-Day		
		Observed Concentration (Mean ± SD, ng/mL)	Precision (%)	Accuracy (%)	Observed Concentration (Mean ± SD, ng/mL)	Precision (%)	Accuracy (%)
plasma	10	9.56 ± 1.02	10.64	95.62	10.85 ± 1.11	10.22	108.51
	200	194.63 ± 9.95	5.11	97.32	209.76 ± 19.05	9.08	104.88
	2000	2081.86 ± 82.42	3.96	104.09	1978.55 ± 35.45	6.85	98.93
liver	10	9.45 ± 1.05	11.09	94.47	-	-	-
	200	196.68 ± 9.46	8.77	97.36	-	-	-
	2000	2032.81 ± 11.12	5.22	102.58	-	-	-
Intestine tract	10	9.24 ± 0.39	9.38	95.80	-	-	-
	200	197.92 ± 13.76	6.85	104.21	-	-	-
	2000	2047.83 ± 27.39	1.34	102.39	-	-	-

**Table 3 ijms-25-00835-t003:** Recovery and matrix effect of RT4 in the UC mice plasma.

Nominal Concentration (ng/mL)	Recovery Effect (%)		Matrix Effect (%)	
	Mean ± SD	RSD	Mean ± SD	RSD
10	91.02 ± 8.52	9.36	93.42 ± 5.24	5.60
200	92.51 ± 1.99	2.15	93.03 ± 4.80	5.15
2000	96.58 ± 6.81	7.05	97.67 ± 5.98	6.12

**Table 4 ijms-25-00835-t004:** Stability of RT4 in the UC mice plasma.

Level	QC_L_		QC_H_	
Stability	Precision (%)	Accuracy (%)	Precision (%)	Accuracy (%)
Storage at 25 ± 2 °C for 24 h	9.57	94.88	6.80	101.47
Storage at −80 °C for 1 month	9.33	88.35	6.33	99.47
Three freeze-thaw cycles	7.08	96.85	3.23	94.43
4 °C Auto-sampler 12 h	7.45	91.13	5.49	92.96

**Table 5 ijms-25-00835-t005:** Main pharmacokinetic parameters of RT4 in the UC mice.

	RT4-ig-L (10 mg/kg)	RT4-ig-M (20 mg/kg)	RT4-ig-H (40 mg/kg)	RT4-iv (2 mg/kg)
*AUC*_0-t_ (μg·h/L)	991.55 ± 216.60	2438.90 ± 568.77	5288.20 ± 1787.98	1246.37 ± 352.57
*AUC*_0-∞_ (μg·h/L)	1139.92 ± 229.03	2698.59 ± 712.03	5993.63 ± 1988.35	1380.43 ± 267.73
*t*_1/2_ (h)	4.67 ± 1.34	6.29 ± 2.18	7.17 ± 2.78	2.24 ± 0.72
*T*_max_ (h)	1.50 ± 1.22	1.17 ± 0.41	1.33 ± 0.52	0.03 ± 0.00
*V*_d_ (L/kg)	61.91 ± 26.25	71.24 ± 31.35	71.60 ± 24.78	4.57 ± 0.82
*CL* (L/h/kg)	9.12 ± 2.12	8.04 ± 2.95	7.20 ± 1.97	1.51 ± 0.40
*C*_max_ (μg/L)	165.13 ± 46.75	295.59 ± 74.25	658.65 ± 264.01	1970.34 ± 486.78

## Data Availability

Data are contained within the article.

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
