# Peer review of "Study on the Anti-Ulcerative Colitis Effect of Pseudo-Ginsenoside RT4 Based on Gut Microbiota, Pharmacokinetics, and Tissue Distribution"

_ijms, 2024, doi:10.3390/ijms25020835_

Round 1
Reviewer 1 Report
Comments and Suggestions for Authors
Reviewer comments and suggestions
The authors in this study explored the therapeutic effect of oral administration of pseudo-ginsenoside RT4 (RT4) on ulcerative colitis (UC), and to determine the absorption and distribution in mice.
Balb/c mice were induced by dextran sulfate sodium salt (DSS) to establish the UC model and were administered by gavage with 10, 20, and 40 mg/kg of RT4.
Caco-2 cells were stimulated by LPS to create an epithelial barrier damage model before being treated with 4, 8, or 16 g/ml of RT4. Finally, pharmacokinetic and tissue distribution experiments in UC mice after intragastric injection of RT4 were carried out. The study's findings in mice were as follows: RT4 improved colonic histopathology, preserved the ultrastructure of colon mucosal epithelial cells, corrected disruption of intestinal microbiota, decreased the DAI score, restored colon length, and decreased levels of pro-inflammatory cytokines (TNF-α, IL-6, and IL-1β). It also increased levels of the immunosuppressive cytokine IL-10.
Additionally, the main pharmacokinetic parameters (Cmax, Tmax,t1/2, Vd, CL, AUC) were obtained, and the main distribution tissues were the small intestine and colon. Hence, the authors suggested that RT4 have characteristics of slow elimination and directional distribution that could alleviate UC by inhibiting inflammatory factors, repairing the intestinal mucosal barrier, and boosting the dominant intestinal microflora.
Overall, the manuscript was good. However, a few major concerns/comments needed to be explained or modified.
- Lines 49-50 Please mention the nature of these compounds for the common reader of your manuscript
- Line 59-60 Which study, please cite
- Line 75-77 These sentences are not required in my view
- Line 102 Please explain the relationship
- Line 241 (2.1.7)The name of the section was similar to the previous one
- Line 348 and 354 Is this small subsection should be differentiate/ if not you can merge
- These are general parts that the authors mention in the first paragraph of the discussion. It would be nice if they could add the novelty in the first paragraph.
- Line 430-432 please add up the table or figures to support your points.
- Line 456-459 It would be nice if the authors cite previous publications that may relate to his or her study. In this way, the discussion section may enrich, and also reader could follow other findings that relate to your study.
- Line 467 What does the result indicate that needs to be mentioned?
- Line 507 Please explain it in one line as well
- Please check the journal list as I found that the names of the journals were in capital format. Please modify it with small letters.
Author Response
|
Response to Reviewer 1 Comments |
||
|
1. Summary |
|
|
|
We sincerely appreciate all the constructive comments by you. Based on the comments, the manuscript has been carefully revised. All the changes to the text have been marked in red in the revised manuscript. The responses to the comments are listed below. In addition, the manuscript has been thoroughly checked before submission. |
||
|
2. Questions for General Evaluation |
Reviewer’s Evaluation |
Response and Revisions |
|
Does the introduction provide sufficient background and include all relevant references? |
Can be improved |
|
|
Are all the cited references relevant to the research? |
Yes |
|
|
Is the research design appropriate? |
Yes |
|
|
Are the methods adequately described? |
Yes |
|
|
Are the results clearly presented? |
Can be improved |
|
|
Are the conclusions supported by the results? |
Yes |
|
|
3. Point-by-point response to Comments and Suggestions for Authors |
||
|
Comments 1: Lines 49-50 Please mention the nature of these compounds for the common reader of your manuscript |
||
|
Response 1: Thank you for pointing this out. We agree with this comment. Therefore, we have We have supplemented the nature of these compounds in the revised manuscript as follows: “M10, a naturally occurring flavonoid of myricetin derivatized with the properties of colon-specific distribution, was reported as treating UC mice by modifying the composition of the gut microbiota. Puerarin, a natural polyphenol product, may relieve UC via enhancing mucin secretion, regulating mucin-utilizing bacteria, and increasing levels of SCFAs in mice. Artesunate, a diterpenoid lactone, has also shown positive anti-UC effects through significantly protecting the integrity of the intestinal barrier and reducing inflammatory cell infiltration in UC mice. Ginsenoside Rh1, a tetracyclic triterpenoid glycoside, has also shown the regulatoy activity in relation to the crosstalk between macrophage and gut microbiota in UC mice.”(page 2, paragraph 1, line 51-58) |
||
|
Comments 2: Line 59-60 Which study, please cite |
||
|
Response 2: Thank you for pointing this out. We have added references in the revised manuscript. (page 2, paragraph 1, line 67) |
||
|
Comment 3: Line 75-77 These sentences are not required in my view |
||
|
Response 3: We have deleted these sentences in the revised manuscript. |
||
|
Comment 4: Line 102 Please explain the relationship |
||
|
Response 4: Thank you for pointing this out. We have supplemented this relationship in the revised manuscript as follows: “The degree of splenomegaly was also positively correlated with the severity of colitis. The higher the degree of splenomegaly, the more extensive the development of colitis. ”(page 3, paragraph 3, line 112-113) |
||
|
Comment 5: Line 241 (2.1.7)The name of the section was similar to the previous one |
||
|
Response 5: Thank you for pointing this out. We have revised this in the the revised manuscript as follows: “2.1.7. Correlation between Gut Microbiota and UC Evaluating Indicators”(page 7, line 255) |
||
|
Comment 6: Line 348 and 354 Is this small subsection should be differentiate/ if not you can merge |
||
|
Response 6: Thank you for pointing this out. Dilution reliability is demonstrated by adding analytes to the matrix to a concentration higher than the upper limit of quantification and diluting the sample with a blank matrix. Residues are estimated by injecting blank samples after injecting high concentration samples or calibration standards. So, these two subsections are differentiate. |
||
|
Comment 7: These are general parts that the authors mention in the first paragraph of the discussion. It would be nice if they could add the novelty in the first paragraph. |
||
|
Response 7: Thank you for pointing this out. We have added the novelty of RT4 in the discussion part in the revised manuscript.(page 17, paragraph 1, line 468-470) |
||
|
Comment 8: Line 430-432 please add up the table or figures to support your Comments. |
||
|
Response 8: Thank you for pointing this out. We have added a figure in the discussion part to summarize the role of RT4 in the treatment of ulcerative colitis. (page 19, line 560-561) |
||
|
Comment 9: Line 456-459 lt would be nice if the authors cite previous publications that may relate to his or her study. ln this way, the discussion section may enrich, and also reader could follow other findings that relate to your study. |
||
|
Response 9: Thank you for pointing this out. We have added references in the revised manuscript. (page 18, paragraph 2, line 501) |
||
|
Comment 10: Line 467 What does the result indicate that needs to be mentioned? |
||
|
Response 10: Thank you for pointing this out. We have supplemented conclusions in the revised manuscript as follows: “Which indicated that RT4 could be absorbed into the blood after oral administration to UC mice, the maximum plasma concentration could be reached in 1~1.5 h, and the plasma concentration could be maintained at a certain level within 5~7 h. This is closely related to the therapeutic effect of RT4 for UC.”(page 18, paragraph 5, line 544-549) |
||
|
Comment 11: Line 507 Please explain it in one line as well |
||
|
Response 11: Thank you for pointing this out. We have adjusted this sentence as follows: “And the disease activity index (DAI) score represents one-third of the total of the above three scores.”(page 19, paragraph 3, line 600-601) |
||
|
Comment 12: Please check the journal list as I found that the names of the journals were in capital format. Please modify it with small letters. |
||
|
Response 12: Thank you for pointing this out. We have corrected this problem in the revised manuscript. |
||
Reviewer 2 Report
Comments and Suggestions for Authors
This manuscript delves into the therapeutic effects of orally administered pseudo-ginsenoside RT4 (RT4) on ulcerative colitis (UC) using a DSS-induced mice model and an LPS-induced Caco-2 cell model for epithelial barrier damage. The author endeavors to demonstrate that RT4 can mitigate UC by inhibiting inflammatory factors, restoring the intestinal mucosal barrier, and modulating the interaction of intestinal microflora along with the levels of short-chain fatty acids (SCFAs). While previous studies have reported the protective effects of RT4 on UC, this manuscript provides a novel perspective, backed by extensive research, to substantiate the therapeutic potential of RT4 in UC. The findings hold significance for future drug development.
However, there are several points that warrant attention. (1) clarification is needed on the percentage of RT4 purity. (2) the author attempts to elucidate the superiority of oral administration over intraperitoneal injection, and supporting data should be gathered. (3) initial abbreviations should be accompanied by their full names, as indicated in line 17. Additionally, certain expressions need correction. For instance, in Figure 2, the results pertain to the effect of RT4 on inflammation, not SCFAs. No SCFAs results are presented in this section (Line 134). (4) the manuscript should delve into the interrelation among microbiota abundance, SCFAs, and inflammation. What is the primary reason for RT4's effect on UC? This relationship requires further exploration. (5) the association analysis between specific intestinal bacteria and inflammatory factors, while conducted, may be insufficient to verify the impact of microbiota on inflammation in UC. Utilizing both cell models and mice models is recommended to corroborate the effect of specific microbiota on inflammation.
Author Response
|
Response to Reviewer 2 Comments |
||
|
1. Summary |
|
|
|
We sincerely appreciate all the constructive comments by you. Based on the comments, the manuscript has been carefully revised. All the changes to the text have been marked in red in the revised manuscript. The responses to the comments are listed below. In addition, the manuscript has been thoroughly checked before submission. |
||
|
2. Questions for General Evaluation |
Reviewer’s Evaluation |
Response and Revisions |
|
Does the introduction provide sufficient background and include all relevant references? |
Yes |
|
|
Are all the cited references relevant to the research? |
Yes |
|
|
Is the research design appropriate? |
Can be improved |
|
|
Are the methods adequately described? |
Yes |
|
|
Are the results clearly presented? |
Can be improved |
|
|
Are the conclusions supported by the results? |
Can be improved |
|
|
3. Point-by-point response to Comments and Suggestions for Authors |
||
|
Comments 1: clarification is needed on the percentage of RT4 purity. |
||
|
Response 1: Thank you for pointing this out. We have supplemented it in the revised manuscript. (page 19, paragraph 3, line 565) |
||
|
Comments 2: the author attempts to elucidate the superiority of oral administration over intraperitoneal injection, and supporting data should be gathered. |
||
|
Response 2: Thank you for pointing this out. Supporting data has been gathered based on the compared our research with the literature report, and the supporting data has been added in the revised version as follows: “The superiority of oral administration over intraperitoneal injection can be elucidated as follows: First of all, oral administration is more convenient with no directly damage to the skin and mucosa, and patients displayed better rates of compliance with the drug regimen. Additionally, in terms of weight, DAI score, colon length, and inflammatory factors (TNF-α, IL-1β, IL-10), the effects of oral administration of RT4 are basically consistent, compared with the pharmacological effects of intraperitoneal injection of RT4 reported in literature. Moreover, administration of RT4 by gavage could selectively enrich in the colon, which could target to the treatment of the ulcerative colitis.”(page 17, paragraph 1, line 470-478) |
||
|
Comment 3: initial abbreviations should be accompanied by their full names, as indicated in line 17. Additionally, certain expressions need correction. For instance, in Figure 2, the results pertain to the effect of RT4 on inflammation, not SCFAs. No SCFAs results were presented in this section (Line 134). |
||
|
Response 3: Thank you for pointing this out. We have added the full names of SCFAs in line 17. (page 1, line 18) In addition, we have adjusted the position of the figure from section 2.1.2 to section 2.1.3. (page 4, line 157-158) |
||
|
Comment 4: the manuscript should delve into the interrelation among microbiota abundance, SCFAs, and inflammation. What is the primary reason for RT4's effect on UC? This relationship requires further exploration. |
||
|
Response 4: Thank you for pointing this out. The relationship between microbiota abundance, SCFAs, and inflammation has been further explored as follows: “Overall, the primary reason for RT4's effect on UC is as follows: “under the intervention of RT4, the gut microbiota were regulated, leading to an increase in the content of metabolites SCFAs (serving as energy sources for colonic cells and anti-inflammatory agents), a decrease in the secretion of inflammatory factors caused by gut microbiota, a relief in of the imbalance between inflammatory and anti-inflammatory factors, a reduction in sustained inflammation of the intestinal mucosa, and a reduction in intestinal mucosal barrier damage.”(page 18, paragraph 4, line 533-539) |
||
|
Comment 5: the association analysis between specific intestinal bacteria and inflammatory factors, while conducted, may be insufficient to verify the impact of microbiota on inflammation in UC. Utilizing both cell models and mice models is recommended to corroborate the effect of specific microbiota on inflammation. |
||
|
Response 5: Thank you for pointing this out. It is true that the association analysis between specific intestinal bacteria and inflammatory factors has not been conducted in our current study. But the association between them in UC patients has been reported, such as acultative anaerobic bacteria were increased and the obligately anaerobic producers of SCFAs were decreased[32]. In our study, there was an increase in facultative anaerobic bacteria (such as Escherichia-Shigella) , and some obligately anaerobic producers (such as Alistipes, Alloprevotella, the Rikenellaceae RC9 gut group) related to SCFAs were relatively reduced in UC mice, which was consistent with the microbial community in UC patients. These microbiota do indeed have an effect on ulcerative colitis, and the function of specific intestinal bacteria has been validated in UC patients. Additionally, there were normal control group, model group, and high dose of RT4 group in our Gut Microbiota analysis. These specific intestinal bacteria were screened based on the levels in the above three groups, namely, they were changed in model group compared with normal control group, and they were re-regulated in RT4 group. The screening could verify the impact of microbiota on inflammation in UC. And there were also some similar research with the same research ideas with us [1-3]. Overall, in our opinion, it is sufficient to verify the impact of microbiota on inflammation in UC in our current study. But it is indeed a good suggestion to utilize both cell models and mice models to corroborate the effect of specific microbiota on inflammation, and we will conduct the study in our future work. (page 18, paragraph 1, line498-504)
[1] Xu, D.w.; Zhuang, L.; Gao, S.; Ma, H.; Cheng, J.; Liu, J.x.; Liu, D.F.; Fu, S.P.; Hu, G.Q. Orally Administered Ginkgolide C Attenuates DSS-Induced Colitis by Maintaining Gut Barrier Integrity, Inhibiting Inflammatory Responses, and Regulating Intestinal Flora. J. Agric. Food Chem. 2022, 70, 14718-14731. [2] Zou, Q.h.; Zhang, X.; Liu, X.H.;Li, Y.T.; Tan Q.L.; Dan, Q.; Yuan, T.; Liu, X.B.; Liu, R.H.; Liu, Z.G. Ficus carica polysaccharide attenuates DSS-induced ulcerative colitis in C57BL/6 mice. Food Funct. 2020, 11, 6666-6679. [3] Hu, Q.j.; Yu, L.l.; Zhai, Q.X.; Zhao, J.X.; Tian, F.W. Anti-Inflammatory, Barrier Maintenance, and Gut Microbiome Modulation Effects of Saccharomyces cerevisiae QHNLD8L1 on DSS-Induced Ulcerative Colitis in Mice. Int. J. Mol. Sci. 2023, 24, 6721.
|
||
Reviewer 3 Report
Comments and Suggestions for Authors
The study aimed to investigate the therapeutic effects of orally administered pseudo-ginsenoside RT4 (RT4) on ulcerative colitis (UC) in Balb/c mice induced by dextran sulfate sodium salt (DSS). It also sought to determine the absorption and distribution of RT4 in mice with UC. The results indicated that RT4 had positive effects on clinical symptoms, inflammatory responses, intestinal barrier function, short-chain fatty acids (SCFAs) content, and gut microbiota. The study included experiments on Caco-2 cells to evaluate the impact on epithelial barrier damage. Pharmacokinetic and tissue distribution studies were conducted after intragastric administration of RT4 in UC mice.
In general, I have no comments or suggestions for this manuscript.
Author Response
|
Response to Reviewer 3 Comments |
||
|
1. Summary |
|
|
|
Thanks so much for your recognition of our work. |
||
|
2. Questions for General Evaluation |
Reviewer’s Evaluation |
Response and Revisions |
|
Does the introduction provide sufficient background and include all relevant references? |
Yes |
|
|
Are all the cited references relevant to the research? |
Yes |
|
|
Is the research design appropriate? |
Yes |
|
|
Are the methods adequately described? |
Yes |
|
|
Are the results clearly presented? |
Yes |
|
|
Are the conclusions supported by the results? |
Yes |
|
Reviewer 4 Report
Comments and Suggestions for Authors
The manuscript by Yu et al. explores the therapeutic potential of pseudoginsenoside RT4 (RT4) in treating ulcerative colitis (UC), a chronic inflammatory condition of the colon. The study's focus is on the oral administration of RT4 and its impact on UC symptoms, inflammatory responses, the intestinal barrier, and gut microbiota in mice. Additionally, the pharmacokinetics and tissue distribution of RT4 following intragastric administration in UC-induced mice are investigated. The study utilizes Balb/c mice with dextran sulfate sodium salt-induced UC and Caco-2 cells to demonstrate the efficacy of RT4. Key findings include the reduction in disease activity index (DAI), restoration of colon length, decrease in pro-inflammatory cytokines, and increase in anti-inflammatory cytokine levels. Additionally, RT4 was shown to improve colonic histopathology and the ultrastructure of colon mucosal epithelial cells, along with modulating the composition of gut microbiota and increasing short-chain fatty acids (SCFAs) content. The comprehensive analysis of its effects on UC symptoms, gut microbiota, and intestinal barrier integrity, combined with pharmacokinetic studies, presents a compelling case for further exploration of RT4 as a therapeutic agent. This research paper looks interesting to me. I am quite open to looking at a revised version if the authors could address some major and minor issues in a satisfactory fashion, which I describe in more detail below.
Major issues:
1. The authors need to specify what statistical tests are used for figures and results. For instance, they mentioned the p-value in line 93 and Figure 1&2 for the first time, but they did not specify what statistical tests were used. Did they use the Dunne’s t-test they wrote in the Methods section? Please add more details so that readers can understand it better.
2. In Figure 7, there are weird tilted red lines below texts such as Occludi or E-cadherin. Please remove them.
3. In Figure 8, the figure caption is too concise to understand the technical details of (Multiple Reaction Monitoring) MRM chromatograms. Please include more details regarding how to read those MRM chromatograms so that readers can better comprehend it.
4. While the study presents significant findings, the relevance and translatability of these results to human UC should be discussed, given the inherent differences between human and mouse physiology.
5. The paper could benefit from a more detailed discussion comparing RT4's efficacy with current standard treatments for UC, providing a clearer context for its potential use in clinical settings.
6. It would be helpful to include future research directions, particularly focusing on the long-term effects and safety profile of RT4 in prolonged use scenarios.
7. I suggest a better description of the relationships between gut microbes and metabolites in the discussion. Although the authors summarize some literature evidence of interactions between gut microbes and metabolites such as SCFAs (Short-Chain Fatty Acids), there are some works that have systematically captured their interactions that have been found in many pieces of literature (Akshit Goyal et al., Nature Communications 2021; Jaeyun Sung et al., Nature Communications 2017). Therefore, the authors need to properly summarize existing papers.
8. There are many grammatical errors in the manuscript that need to be fixed. Some grammatical modifications are suggested below in my minor comments. Please check the entire manuscript and fix all errors, specifically the errors related to definite and indefinite articles.
Minor comments:
1. Line 27: “proteins expression” -> “protein expression”
2. Line 31: “, modulating” -> “, and modulating”
3. Lines 36-37: “with growing global morbidity rate” -> “with a growing global morbidity rate”
4. Line 43: “long time use” -> “long-time use”
5. Line 49: “by modifying composition” -> “by modifying the composition”
6. Line 60: “improve pathological state” -> “improve the pathological state”
7. Line 81: “UC model” -> “the UC model”
8. Line 93: “high dose” -> “a high dose”
9. Line 98: “were” -> “was”
10. Line 129: “markably” -> “markedly”
11. Line 165: “insight on” -> “insight into”
12. Line 173: “there were almost no degeneration of organelles” -> “there was almost no degeneration of organelles”
13. Line 192: “the shannon index” -> “the Shannon index”
14. Line 281: “obvious higher” -> “obviously higher”
15. Line 388: “It was showed” -> “It was shown”
16. Line 420: “once withdraw” -> “once withdrawn.”
17. Lines 470-471: “and the models established in current research is tend to be acute inflammation” should be “and the models established in the current research tend to be of acute inflammation.”
18. Lines 490-491: “Comply with the ‘Guide for the Care and Use of Laboratory Animals’” should be “Complying with the ‘Guide for the Care and Use of Laboratory Animals’.”
19. Line 505: “Body Weigh” -> “Body Weight”
20. Line 518: “Then the colon were rinsed” should be “Then the colon was rinsed.”
21. Line 519: “Part of the colon tissue was used” should be “A part of the colon tissue was used.”
22. Lines 542-543: “was accurately weighed and was then homogenized” should be “was accurately weighed and then homogenized.”
23. Line 671: “The bioanalytical method validation were carried on” should be “The bioanalytical method validation was carried out.”
24. Line 739: “In current study” -> “In the current study”
25. Lines 742-743: “coupled by” -> “coupled with”
Comments on the Quality of English LanguageThere are many grammatical errors in the manuscript that need to be fixed. Some grammatical modifications are suggested below in my minor comments. Please check the entire manuscript and fix all errors, specifically the errors related to definite and indefinite articles.
Author Response
|
Response to Reviewer 4 Comments |
||
|
1. Summary |
|
|
|
We sincerely appreciate all the constructive comments by you. Based on the comments, the manuscript has been carefully revised. All the changes to the text have been marked in red in the revised manuscript. The responses to the comments are listed below. In addition, the manuscript has been thoroughly checked before submission. |
||
|
2. Questions for General Evaluation |
Reviewer’s Evaluation |
Response and Revisions |
|
Does the introduction provide sufficient background and include all relevant references? |
Yes |
|
|
Are all the cited references relevant to the research? |
Yes |
|
|
Is the research design appropriate? |
Yes |
|
|
Are the methods adequately described? |
Can be improved |
|
|
Are the results clearly presented? |
Must be improved |
|
|
Are the conclusions supported by the results? |
Can be improved |
|
|
3. Point-by-point response to Comments and Suggestions for Authors |
||
|
Comments 1: The authors need to specify what statistical tests are used for figures and results. For instance, they mentioned the p-value in line 93 and Figure 1&2 for the first time, but they did not specify what statistical tests were used. Did they use the Dunne’s t-test they wrote in the Methods section? Please add more details so that readers can understand it better. |
||
|
Response 1: Thank you for pointing this out. Yes, we did use the Dunne’s t-test, we have supplemented it in the Figure 1&2 in the revised manuscript. (page 3, line 126; page 4, line 161) |
||
|
Comments 2: In Figure 7, there are weird tilted red lines below texts such as Occludi or E-cadherin. Please remove them. |
||
|
Response 2: Thank you for pointing this out. These weird tilted red lines are spell checking features in WPS, We have removed tilted red lines below texts in the revised manuscript. (page 11, line 324) |
||
|
Comment 3: In Figure 8, the figure caption is too concise to understand the technical details of (Multiple Reaction Monitoring) MRM chromatograms. Please include more details regarding how to read those MRM chromatograms so that readers can better comprehend it. |
||
|
Response 3: Thank you for pointing this out. We have added more details in the Figure 8 caption as follows: “Figure 8. MRM chromatograms of RT4 (RT: 0.58 min; m/z: 655.4→143.5) and PD (RT: 3.68 min; m/z: 461.1→425.1).” (page 4, line 338-339) |
||
|
Comment 4: While the study presents significant findings, the relevance and translatability of these results to human UC should be discussed, given the inherent differences between human and mouse physiology. |
||
|
Response 4: Thank you for pointing this out. The relevance and translatability of the results to human UC has been discussed as follows: “Patients with UC have many clinical manifestations. Hemorrhagic diarrhea is the most common early symptom. Other symptoms include abdominal pain, bloody stools, weight loss, urgency and heaviness, and vomiting, etc. The common diagnostic tests include blood samples and colonoscopies. The etiology and pathogenesis of UC are related to the interaction of multiple factors such as environmental factors, genetic factors, and gut microbiota, leading to an intestinal immune imbalance (imbalance of inflammatory/ anti-inflammatory factors). In the current study, the therapeutic effects of orally administered RT4 on UC in Balb/c mice induced by DSS were investigated. Key findings included the reduction in DAI, the restoration of colon length, decrease in pro-inflammatory cytokines, an increase in anti-inflammatory cytokine levels, improvement in the colonic histopathology and ultrastructure, modulation of the composition of gut microbiota, and an increase in the content of SCFAs. Except for colon length, the DAI, inflammatory/anti-inflammatory factors, colonic histopathology, gut microbiota, and the content of SCFAs all have the relevance and translatability to human UC.” (page 18, paragraph 3, line 518-532) |
||
|
Comment 5: The paper could benefit from a more detailed discussion comparing RT4's efficacy with current standard treatments for UC, providing a clearer context for its potential use in clinical settings. |
||
|
Response 5: Thank you for pointing this out. We have compared the RT4's efficacy with current standard treatments for UC in the revised manuscript as follows: “In general, compared with the current clinical drugs for UC, RT4 has the characteristics of rapid absorption and slow elimination, and could be specifically distributed in colonic tissue.” (page 18, paragraph 5, line 547-549) |
||
|
Comment 6: It would be helpful to include future research directions, particularly focusing on the long-term effects and safety profile of RT4 in prolonged use scenarios. |
||
|
Response 6: Thank you for pointing this out. We have added the long-term effects and safety profile of RT4 in the the revised manuscript as follows: “In further studies, an evaluation of its safety and toxicity should be a priority.”(page 19, paragraph 2, line 556-559) |
||
|
Comment 7: I suggest a better description of the relationships between gut microbes and metabolites in the discussion. Although the authors summarize some literature evidence of interactions between gut microbes and metabolites such as SCFAs (Short-Chain Fatty Acids), there are some works that have systematically captured their interactions that have been found in many pieces of literature (Akshit Goyal et al., Nature Communications 2021; Jaeyun Sung et al., Nature Communications 2017). Therefore, the authors need to properly summarize existing papers. |
||
|
Response 7: Thank you for pointing this out. We have added a better description of the relationships between gut microbes and metabolites in the discussion as follows: “There is a large network of cross-feeding interactions between the gut microbiome and the metabolites they produce.”(page 17, paragraph 1, line 445-447) |
||
|
Comment 8: There are many grammatical errors in the manuscript that need to be fixed. Some grammatical modifications are suggested below in my minor comments. Please check the entire manuscript and fix all errors, specifically the errors related to definite and indefinite articles. Minor comments: 1. Line 27: “proteins expression” -> “protein expression” 2. Line 31: “, modulating” -> “, and modulating” 3. Lines 36-37: “with growing global morbidity rate” -> “with a growing global morbidity rate” 4. Line 43: “long time use” -> “long-time use” 5. Line 49: “by modifying composition” -> “by modifying the composition” 6. Line 60: “improve pathological state” -> “improve the pathological state” 7. Line 81: “UC model” -> “the UC model” 8. Line 93: “high dose” -> “a high dose” 9. Line 98: “were” -> “was” 10. Line 129: “markably” -> “markedly” 11. Line 165: “insight on” -> “insight into” 12. Line 173: “there were almost no degeneration of organelles” -> “there was almost no degeneration of organelles” 13. Line 192: “the shannon index” -> “the Shannon index” 14. Line 281: “obvious higher” -> “obviously higher” 15. Line 388: “It was showed” -> “It was shown” 16. Line 420: “once withdraw” -> “once withdrawn.” 17. Lines 470-471: “and the models established in current research is tend to be acute inflammation” should be “and the models established in the current research tend to be of acute inflammation.” 18. Lines 490-491: “Comply with the ‘Guide for the Care and Use of Laboratory Animals’” should be “Complying with the ‘Guide for the Care and Use of Laboratory Animals’.” 19. Line 505: “Body Weigh” -> “Body Weight” 20. Line 518: “Then the colon were rinsed” should be “Then the colon was rinsed.” 21. Line 519: “Part of the colon tissue was used” should be “A part of the colon tissue was used.” 22. Lines 542-543: “was accurately weighed and was then homogenized” should be “was accurately weighed and then homogenized.” 23. Line 671: “The bioanalytical method validation were carried on” should be “The bioanalytical method validation was carried out.” 24. Line 739: “In current study” -> “In the current study” 25. Lines 742-743: “coupled by” -> “coupled with” |
||
|
Response 8: Thank you for pointing this out. We have corrected the above mentioned grammatical errors in the revised manuscript and also further checked for other grammatical errors in the manuscript. |
||
Round 2
Reviewer 4 Report
Comments and Suggestions for Authors
The authors have provided satisfactory responses to all of my inquiries. I have no additional comments to make.